# On the Similarity between the Laplace and Neural Tangent Kernels

**Amnon Geifman**[1]          **Abhay Yadav**[2]          **Yoni Kasten**[1]

**Meirav Galun**[1]          **David Jacobs**[2]          **Ronen Basri**[1]

[1]Department of Computer Science, Weizmann Institute of Science, Rehovot, Israel
`{amnon.geifman, yoni.kasten, meirav.galun, ronen.basri}@weizmann.ac.il`

[2]Department of Computer Science, University of Maryland, College Park, MD
`{jaiabhay,djacobs}@cs.umd.edu`

## Abstract

Recent theoretical work has shown that massively overparameterized neural networks are equivalent to kernel regressors that use *Neural Tangent Kernels* (NTKs). Experiments show that these kernel methods perform similarly to real neural networks. Here we show that NTK for fully connected networks with ReLU activation is closely related to the standard Laplace kernel. We show theoretically that for normalized data on the hypersphere both kernels have the same eigenfunctions and their eigenvalues decay polynomially at the same rate, implying that their Reproducing Kernel Hilbert Spaces (RKHS) include the same sets of functions. This means that both kernels give rise to classes of functions with the same smoothness properties. The two kernels differ for data off the hypersphere, but experiments indicate that when data is properly normalized these differences are not significant. Finally, we provide experiments on real data comparing NTK and the Laplace kernel, along with a larger class of $\gamma$-exponential kernels. We show that these perform almost identically. Our results suggest that much insight about neural networks can be obtained from analysis of the well-known Laplace kernel, which has a simple closed form.

## 1 Introduction

Neural networks with significantly more parameters than training examples have been successfully applied to a variety of tasks. Somewhat contrary to common wisdom, these models typically generalize well to unseen data. It has been shown that in the limit of infinite model size, these neural networks are equivalent to kernel regression using a family of novel *Neural Tangent Kernels* (NTK) [26, 2]. NTK methods can be analyzed to explain many properties of neural networks in this limit, including their convergence in training and ability to generalize [8, 9, 13, 32]. Recent experimental work has shown that in practice, kernel methods using NTK perform similarly, and in some cases better, than neural networks [4], and that NTK can be used to accurately predict the dynamics of neural networks [1, 2, 7]. This suggests that a better understanding of NTK can lead to new ways to analyze neural networks.

These results raise an important question: Is NTK significantly different from standard kernels? For the case of fully connected (FC) networks, [4] provides experimental evidence that NTK is especially effective, showing that it outperforms the Gaussian kernel on a large suite of machine learning problems. Consequently, they argue that NTK should be added to the standard machine learning

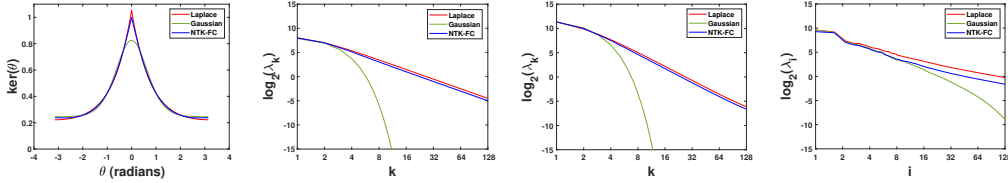

Figure 1: Left: An overlay of the NTK for a 6-layer FC network with ReLU activation with the Laplace and Gaussian kernels, as a function of the angle between their arguments. The exponential kernels are modulated by an affine transformation to achieve a least squares fit to the NTK. Note the high degree of similarity between the Laplace kernel and NTK. Middle left: eigenvalues as a function of frequency in $\mathbb{S}^1$. The slopes in these log-log plots indicate the rate of decay, which is similar for both the Laplace kernel and for NTK for the FC network with 6 layers. (Empirical slopes are -1.94 for both Laplace and NTK-FC.) The eigenvalues of the Gaussian kernel, in contrast, decay exponentially. Middle right: Same for $\mathbb{S}^2$. (Empirical slopes are -2.75 for the Laplace and NTK-FC.) Right: Same estimated for the UCI Abalone dataset (here we show eigenvalues as function of eigenvalue index).

toolbox. [9] has shown empirically that the dynamics of neural networks on randomly labeled data more closely resembles the dynamics of learning through stochastic gradient descent with the Laplace kernel than with the Gaussian kernel. In this paper we show theoretically and experimentally that NTK does closely resemble the Laplace kernel, already a standard tool of machine learning.

Kernels are mainly characterized by their corresponding Reproducing Kernel Hilbert Space (RKHS), which determines the set of functions they can produce [28]. They are further characterized by the RKHS norm they induce, which is minimized (implicitly) in every regression problem. Our main result is that when restricted to the hypersphere $\mathbb{S}^{d-1}$, NTK for a fully connected (FC) network with ReLU activation and bias has the same RKHS as the Laplace kernel, defined as $k^{\mathrm{Lap}}(\mathbf{x}, \mathbf{z}) = e^{-c\|\mathbf{x}-\mathbf{z}\|}$ for points $\mathbf{x}, \mathbf{z} \in \mathbb{S}^{d-1}$ and constant $c > 0$. (In general, NTK for deeper networks is more sharply peaked, corresponding to larger values of $c$, see supplementary material.) This equivalence of RKHSs is shown by establishing that on the hypersphere the eigenfunctions of NTK and the Laplace kernels coincide and their eigenvalues decay at the same rate (see Figure 1), implying in turn that gradient descent (GD) with both kernels should have the same dynamics, explaining [9]'s experiments. In previous work, the eigenfunctions and eigenvalues of NTK have been derived on the hypersphere for networks with only one hidden layer, while these properties of the Laplace kernel have been studied in $\mathbb{R}^d$. We derive new results for the Laplace kernel on the hypersphere, and for NTK for deep networks on the hypersphere and in $\mathbb{R}^d$. In $\mathbb{R}^d$, NTK gives rise to radial eigenfunctions, forgoing the shift invariance property of exponential kernels. Experiments indicate that this difference is not significant in practice.

Finally, we show experiments indicating that the Laplace kernel achieves similar results to those obtained with NTK on real-world problems. We further show that by using the more general, $\gamma$-exponential kernel [41], which allows for one additional parameter, $k^{\gamma}(\mathbf{x}, \mathbf{z}) = e^{-c\|\mathbf{x}-\mathbf{z}\|^{\gamma}}$, we achieve slightly better performance than NTK on a number of standard datasets.

## 2 Related Works

The connection between neural networks and kernel methods has been investigated for over two decades. Early works have noted the equivalence between neural networks with single hidden layers of infinite width and Gaussian Processes (GP) [45, 39], where GP prior can be used to achieve exact Bayesian inference. Recently [30, 34] have extended the results to deep fully-connected neural networks in which all but the last layer retain their initial values. In this context, [17] introduced the Arc-cosine kernel, while [18] showed a duality between neural networks and compositional kernels.

More recent work introduced the family of neural tangent kernels (NTK) [26, 2]. This work showed that for massively overparameterized and fully trained networks, their training dynamics closely follows the path of kernel gradient descent, and that training converges to the solution of kernel regression with NTK. Follow-up work defined analogous kernels for residual [25] and convolutional networks [2, 31]. Recent work also showed empirically that classification with NTK achieves performance similar to deep neural networks with the corresponding architecture [2, 31].

The equivalence between kernels and overparameterized neural networks opened the door to studying inductive bias in neural networks. For two layer, FC networks, [5, 7, 14] investigated the spectral property of the NTK when the data is distributed uniformly on the hypersphere, showing in particular that with GD low frequencies are learned before higher ones. [6] extended these results to non-uniform distributions. [46] analyzed the eigenvalues of NTK over the Boolean cube, and [20] analyzed its spectrum under approximate pairwise orthogonality. [5, 11] further leveraged the spectral properties of the kernels to investigate their RKHS in the case of bias-free two layer networks. Our results apply to deep networks with bias. [24] studied approximation bounds for two layer neural networks, and [8, 9, 13, 32] studied generalization properties of kernel methods in the context of neural networks.

Positive definite kernels and their associated RKHSs have been studied extensively, see, e.g., [43, 44] for reviews. The spectral properties of classic kernels, e.g., the Gaussian and Laplace kernels, are typically derived for input in $\mathbb{R}^d$ [29]. Several papers examine the RKHS of common kernels (e.g., the Gaussian and polynomial) on the hypersphere [36–38].

Recent work compares the performance of NTK to that of common kernels. Specifically, [4]'s experiments suggest that NTK is superior to the Gaussian and low degree polynomial kernels. [9] compares the learning speed of GD for randomly mislabeled data, showing that NTK learns such data as fast as the Laplace kernel and much faster than the Gaussian kernel. Our analysis provides a theoretical justification of this result.

## 3 NTK vs. the Exponential Kernels

Our aim is to compare NTK to common kernels. In comparing kernels we need to consider two main properties: first, what functions are included in their respective RKHS and secondly, how their respective norms behave. (These concepts are reviewed below in Sec. 3.1.) The answer to the former question determines the set of functions considered for regression, while the answer to the latter determines the result of regression. Together, these will determine how a kernel generalizes to unseen data points. Below we see that on the hypersphere both NTK and exponential kernels (e.g., Gaussian and Laplace) give rise to the same set of eigenfunctions. Therefore, the answers to the questions above are determined fully by the corresponding eigenvalues. Moreover, the asymptotic decay rate of the eigenvalues of each kernel determines their RKHS.

As an example consider the exponential kernels in $\mathbb{R}^d$, i.e., the kernels $e^{-c\|\mathbf{x}-\mathbf{z}\|^\gamma}$, where $c > 0$ and $0 < \gamma \le 2$ [41]. These shift invariant kernels have the Fourier transform as their eigenfunctions. The eigenvalues of the Gaussian kernel, i.e., $\gamma = 2$, decay exponentially, implying that its respective RKHS includes only infinitely smooth functions. In contrast, the eigenvalues of the Laplace kernel, i.e., $\gamma = 1$, decay polynomially, forming a space of continuous, but not necessarily smooth functions.

Our main theoretical result is that when restricted to the hypersphere $\mathbb{S}^{d-1}$

$$\mathcal{H}^{\text{Gauss}} \subset \mathcal{H}^{\text{Lap}} = \mathcal{H}^{\text{FC}_\beta(2)} \subseteq \mathcal{H}^{\text{FC}_\beta(\text{L})},$$

where $\mathcal{H}^{\text{Gauss}}$ and $\mathcal{H}^{\text{Lap}}$ denote the RKHSs associated with the Gaussian and Laplace kernels, and $\mathcal{H}^{\text{FC}_\beta(\text{L})}$ denotes the NTK for a FC network with $L$ layers, ReLU activation, and bias. Further empirical results indicate that $\mathcal{H}^{\text{Lap}} = \mathcal{H}^{\text{FC}_\beta(\text{L})}$ for the entire range $L \ge 2$. Indeed, the subsequent work of [16] proves that $\mathcal{H}^{\text{FC}_\beta(2)} \supseteq \mathcal{H}^{\text{FC}_\beta(\text{L})}$, thus together with our results proving that $\mathcal{H}^{\text{FC}_\beta(2)} = \mathcal{H}^{\text{FC}_\beta(\text{L})}$.

Next we briefly recall basic concepts in kernel regression. We subsequently characterize the RKHS of NTK and the Laplace kernel and show their equivalence in $\mathbb{S}^{d-1}$. Finally, we discuss how these kernels extend outside of the sphere to the entire $\mathbb{R}^d$ space. All lemmas and theorems are proved in the supplementary material.

### 3.1 Preliminaries

We consider positive definite kernels $\boldsymbol{k} : \mathcal{X} \times \mathcal{X} \to \mathbb{R}$ defined over a compact metric space $\mathcal{X}$ endowed with a finite Borel measure $\mathcal{V}$. Each such kernel is associated with a Reproducing Kernel Hilbert Space (RKHS) of functions, $\mathcal{H}$, which includes the set of functions the kernel reproduces, i.e., $f(\mathbf{x}) = \langle f, \boldsymbol{k}(\cdot, \mathbf{x}) \rangle_\mathcal{H}$ where the inner product is inherited from the respective Hilbert space. For such kernels the following holds:

1. For all $x \in \mathcal{X}$ we have that the $\boldsymbol{k}(\cdot, x) \in \mathcal{H}$.

2. Reproducing property: for all $x \in \mathcal{X}$ and for all $f \in \mathcal{H}$ it holds that $f(x) = \langle f, \boldsymbol{k}(\cdot, x) \rangle_{\mathcal{H}}$.

Moreover, RKHSs and positive definite kernels are uniquely paired.

According to Mercer's theorem $\boldsymbol{k}$ can be written as

$$\boldsymbol{k}(\mathbf{x}, \mathbf{z}) = \sum_{i \in I} \lambda_i \Phi_i(\mathbf{x}) \Phi_i(\mathbf{z}), \quad \mathbf{x}, \mathbf{z} \in \mathcal{X}, \tag{1}$$

where $\{(\lambda_i, \Phi_i)\}_{i \in I}$ are the eigenvalues and eigenfunctions of $\boldsymbol{k}$ with respect to the measure $\mathcal{V}$, i.e.,

$$\int k(\mathbf{x}, \mathbf{z}) \Phi_i(\mathbf{z}) d\mathcal{V}(\mathbf{z}) = \lambda_i \Phi_i(\mathbf{x}).$$

The RKHS $\mathcal{H}$ is the space of functions $f \in \mathcal{H}$ of the form $f(\mathbf{x}) = \sum_{i \in I} \alpha_i \Phi_i(\mathbf{x})$ whose RKHS norm is finite, i.e., $\|f\|_{\mathcal{H}} = \sum_{i \in I} \frac{\alpha_i^2}{\lambda_i} < \infty$. The latter condition restricts the set of functions in an RKHS, allowing only functions that are sufficiently smooth in accordance with the asymptotic decay of $\lambda_k$.

The literature considers many different kernels (see, e.g., [23]). Here we discuss the family of $\gamma$-exponential kernels $\boldsymbol{k}^{\gamma}(\mathbf{x}, \mathbf{z}) = e^{-c\|\mathbf{x} - \mathbf{z}\|^{\gamma}}$, $0 < \gamma \leq 2$, which include the Laplace ($\gamma = 1$) and the Gaussian ($\gamma = 2$) kernels.

**Neural Tangent Kernel**. Let $f(\theta, \mathbf{x})$ denote a neural network function with ReLU activation and trainable parameters $\theta$. Then the corresponding NTK is defined as

$$\boldsymbol{k}^{\mathrm{NTK}}(\mathbf{x}, \mathbf{z}) = \mathbb{E}_{\theta \sim \mathcal{P}} \left\langle \frac{\partial f(\theta, \mathbf{x})}{\partial \theta}, \frac{\partial f(\theta, \mathbf{z})}{\partial \theta} \right\rangle,$$

where expectation is taken over the probability distribution $\mathcal{P}$ of the initialization of $\theta$, and we assume that the width of each layer tends to infinity. Our results focus on NTK kernels corresponding to deep, fully connected network architectures that may or may not include bias, where bias, if it exists, is initialized at zero. We denote these kernels by $\boldsymbol{k}^{\mathrm{FC}_0(\mathrm{L})}$ for the bias-free version and $\boldsymbol{k}^{\mathrm{FC}_\beta(\mathrm{L})}$ for NTK with bias and define them in the supplementary material.

**Kernel regression**. Given training data $\{(\mathbf{x}_i, y_i)\}_{i=1}^n$, $\mathbf{x}_i \in \mathcal{X}$, $y_i \in \mathbb{R}$, kernel ridge regression is the solution to

$$\min_{f \in \mathcal{H}} \sum_{i=1}^n (f(\mathbf{x}_i) - y_i)^2 + \lambda \|f\|_{\mathcal{H}}^2. \tag{2}$$

When $\lambda \to 0$ this problem is called minimum norm interpolant, and the solution satisfies

$$\min_{f \in \mathcal{H}} \|f\|_{\mathcal{H}} \quad \text{s.t.} \quad \forall i, \ f(\mathbf{x}_i) = y_i. \tag{3}$$

The solution of (2) is given by $f(\mathbf{x}) = \boldsymbol{k}_{\mathbf{x}}^T (K + \lambda I)^{-1} \mathbf{y}$, where the entries of $\boldsymbol{k}_{\mathbf{x}} \in \mathbb{R}^n$ are $\boldsymbol{k}(\mathbf{x}, \mathbf{x}_i)$, $K$ is the $n \times n$ matrix with $K_{ij} = \boldsymbol{k}(\mathbf{x}_i, \mathbf{x}_j)$, $I$ denotes the identity matrix, and $\mathbf{y} = (y_1, ..., y_n)^T$. Further review of kernel methods can be found, e.g., in [28, 43].

## 3.2  NTK in $\mathbb{S}^{d-1}$

We next consider the NTK for fully connected networks applied to data restricted to the hypersphere $\mathbb{S}^{d-1}$. To characterize the kernel, we first aim to determine the eigenvectors of NTK. This will be a direct consequence of Lemma 1. Subsequently in Theorem 1 we will characterize the decay rate of the corresponding eigenvalues.

**Lemma 1.** *Let $\boldsymbol{k}^{\mathrm{FC}_\beta(\mathrm{L})}(\mathbf{x}, \mathbf{z})$, $\mathbf{x}, \mathbf{z} \in \mathbb{S}^{d-1}$, denote the NTK kernels for FC networks with $L \geq 2$ layers, possibly with bias initialized with zero. This kernel is zonal, i.e., $\boldsymbol{k}^{\mathrm{FC}_\beta(\mathrm{L})}(\mathbf{x}, \mathbf{z}) = \boldsymbol{k}^{\mathrm{FC}_\beta(\mathrm{L})}(\mathbf{x}^T \mathbf{z})$. (Note the abuse of notation, which should be clear by context.)*

We note that for the bias-free $\boldsymbol{k}^{\mathrm{FC}_0(\mathrm{L})}$ this lemma was proven in [6] and we extend the proof to allow for bias. It is well known that the spherical harmonics are eigenvectors for any zonal kernel with

respect to the uniform measure on $\mathbb{S}^{d-1}$ with $d \geq 3$. (For background on Spherical Harmonics see, e.g., [22]). Therefore, due to Mercer's Theorem (1), any zonal kernel $\boldsymbol{k}$ can be written as

$$\boldsymbol{k}(\mathbf{x}, \mathbf{z}) = \sum_{k=0}^{\infty} \lambda_k \sum_{j=1}^{N(d,k)} Y_{k,j}(\mathbf{x}) Y_{k,j}(\mathbf{z}), \tag{4}$$

where $Y_{k,j}(.)$ denotes the spherical harmonics of $\mathbb{S}^{d-1}$, $N(d,k)$ denotes the number of harmonics of order $k$ in $\mathbb{S}^{d-1}$, and $\lambda_k$ are the respective eigenvalues. On the circle $\mathbb{S}^1$ the eigenvectors are the Fourier series, and $\boldsymbol{k}(\mathbf{x}, \mathbf{z}) = \sum_{k=0}^{\infty} \frac{1}{c_k} \lambda_k \cos(k\theta)$, where $\theta = \arccos(\mathbf{x}^T \mathbf{z})$ and $c_k$ is a normalization factor, $c_0 = 4\pi^2$ and $c_k = \pi^2$ when $k \geq 1$.

Deriving the eigenvalues for NTK for deep networks is complicated, due to its recursive definition. For a two-layer network without bias, [5, 11] proved that the eigenvalues decay at a rate of $O(k^{-d})$. With no bias, however, two-layer networks are nonuniversal, and in particular $\lambda_k = 0$ for odd $k \geq 3$ [7]. To avoid this issue Theorem 1 establishes that with bias NTK is universal for any number of layers $L \geq 2$, and its eigenvalues decay at a rate no faster than $O(k^{-d})$. Moreover, with $L = 2$ the eigenvalues decay exactly at the rate of $O(k^{-d})$.

**Theorem 1.** *Let* $\mathbf{x}, \mathbf{z} \in \mathbb{S}^{d-1}$. *With bias initialized at zero:*

1. $\boldsymbol{k}^{\mathrm{FC}_\beta(\mathrm{L})}$ *decomposes according to (4) with* $\lambda_k > 0$ *for all* $k \geq 0$, *and*

2. $\exists k_0$ *and constants* $C_1, C_2, C_3 > 0$ *that depend on the dimension* $d$ *such that* $\forall k > k_0$

   (a) $C_1 k^{-d} \leq \lambda_k \leq C_2 k^{-d}$ *if* $L = 2$, *and*
   (b) $C_3 k^{-d} \leq \lambda_k$ *if* $L \geq 3$.

The proof of this theorem for $L = 2$ borrows techniques from [5]. The proof for $L \geq 3$ relies mainly on showing that the algebraic operations in the recursive definition of NTK (including addition, product and composition) do not increase the rate of decay. The consequence of Theorem 1 is that NTK for FC networks gives rise to an infinite size feature space and that its eigenvalues decay no faster than $O(k^{-d})$. While our proofs only establish a bound for the case that $L \geq 3$, empirical results suggest that the eigenvalues for these kernels decay exactly as $\Theta(k^{-d})$, as can be seen in Figure 1.

### 3.3 NTK vs. exponential kernels in $\mathbb{S}^{d-1}$

The polynomial decay of the eigenvalues of NTK suggests that NTK is closely related to the Laplace kernel, as we show next. Indeed, any shift invariant and isotropic kernel, i.e., $\boldsymbol{k}(\mathbf{x}, \mathbf{y}) = \boldsymbol{k}(\|\mathbf{x} - \mathbf{y}\|)$, in $\mathbb{R}^d$ is zonal when restricted to the hypersphere, since $\mathbf{x}, \mathbf{y} \in \mathbb{S}^{d-1}$ implies $\|\mathbf{x} - \mathbf{y}\|^2 = 2(1 - \mathbf{x}^T \mathbf{y})$. Therefore, in $\mathbb{S}^{d-1}$ the spherical harmonics are the eigenvectors of the exponential kernels.

[36] shows that the Gaussian kernel restricted to the hypersphere yields eigenvalues that decay exponentially fast. In contrast we next prove that the eigenvalues of the Laplace kernel restricted to the hypersphere decay polynomially as $\Theta(k^{-d})$, the same decay rate shown for NTK in Theorem 1 and in Figure 1.

**Theorem 2.** *Let* $\mathbf{x}, \mathbf{z} \in \mathbb{S}^{d-1}$ *and write the Laplace kernel as* $\boldsymbol{k}^{\mathrm{Lap}}(\mathbf{x}^T \mathbf{z}) = e^{-c\sqrt{1 - \mathbf{x}^T \mathbf{z}}}$, *restricted to* $\mathbb{S}^{d-1}$. *Then* $\boldsymbol{k}^{\mathrm{Lap}}$ *can be decomposed as in (4) with the eigenvalues* $\lambda_k$ *satisfying* $\lambda_k > 0$, *and* $\exists k_0$ *such that* $\forall k > k_0$ *it holds that:*
$$B_1 k^{-d} \leq \lambda_k \leq B_2 k^{-d}$$
*where* $B_1, B_2 > 0$ *are constants that depend on the dimension* $d$ *and the parameter* $c$.

Our proof uses the decay rate of the Laplace kernel in $\mathbb{R}^d$, and results due to [38, 37] that relate Fourier expansions in $\mathbb{R}^d$ to their corresponding spherical harmonic expansions in $\mathbb{S}^{d-1}$. This allows us to state our main theoretical result.

**Theorem 3.** *Let* $\mathcal{H}^{\mathrm{Lap}}$ *denote the RKHS for the Laplace kernel restricted to* $\mathbb{S}^{d-1}$, *and let* $\mathcal{H}^{\mathrm{FC}_\beta(\mathrm{L})}$ *denote the NTK corresponding respectively to a FC network with* $L$ *layers, ReLU activation, and bias, restricted to* $\mathbb{S}^{d-1}$, *then* $\mathcal{H}^{\mathrm{Lap}} = \mathcal{H}^{\mathrm{FC}_\beta(2)} \subseteq \mathcal{H}^{\mathrm{FC}_\beta(\mathrm{L})}$.

The common decay rates of NTK and the Laplace kernel in $\mathbb{S}^{d-1}$ imply that the set of functions in their RKHSs are identical, having the same smoothness properties. In particular, due to the norm equivalence of RKHSs and Sobolev spaces both spaces include functions that have weak derivatives up to order $d/2$ [38]. We recall that empirical results suggest further that $\boldsymbol{k}^{\mathrm{FC}_\beta(\mathrm{L})}$ decays exactly as $\Theta(k^{-d})$, and so we conjecture that $\mathcal{H}^{\mathrm{Lap}} = \mathcal{H}^{\mathrm{FC}_\beta(\mathrm{L})}$. We note that despite this asymptotic similarity, the eigenvalues of NTK and the Laplace kernel are not identical even if we correct for shift and scale. Consequently, each kernel may behave slightly differently. Our experiments in Section 4 suggest that this results in only small differences in performance.

The similarity between NTK and the Laplace kernel has several implications. First, the dynamics of gradient descent for solving regression (2) with both kernels [15] should be similar. For a kernel with eigenvalues $\{\lambda_i\}_{i=1}^\infty$ a standard calculation shows that GD requires $O(1/\lambda_i)$ time steps to learn the $i$th eigenfunction (e.g., [3, 7]). For both NTK and the Laplace kernel in $\mathbb{S}^{d-1}$ this implies that $O(k^d)$ time steps are needed to learn a harmonic of frequency $k$. This is in contrast for instance with the Gaussian kernel, where the time needed to learn a harmonic of frequency $k$ grows exponentially with $k$. This in particular explains the empirical results of [9], where it was shown that fitting noisy class labels with the Laplace kernel or neural networks requires a similar number of SGD steps. The authors of [9] conjectured that "optimization performance is controlled by the type of non-smoothness," as indeed is determined by the identical RKHS for NTK and the Laplace kernel.

The similarity between NTK and the Laplace kernel also implies that they have similar generalization properties. Indeed various generalization bounds rely explicitly on spectral properties of kernels. For example, given a set of training points $X \subseteq \mathbb{S}^{d-1}$ and a target function $f : \mathbb{S}^{d-1} \to \mathbb{R}$, then the error achieved by the kernel regression estimator given $X$, denoted $\hat{f}_X$, is (see, e.g., [27])

$$\left\| f - \hat{f}_X \right\|_\infty \le C \cdot h(X)^\alpha \|f\|_{\mathcal{H}_k}, \ \ f \in \mathcal{H}_k,$$

where $h(X) := \sup_{\mathbf{z} \in \mathbb{S}^{d-1}} \inf_{\mathbf{x} \in X} \arccos(\mathbf{z}^T \mathbf{x})$ is the mesh norm of $X$ (and thus depends on the density of the points), and $\alpha$ depends on the smoothness property of the kernel. Specifically, for both the Laplace kernel and NTK $\alpha = 1/2$.

Likewise, with $n$ training points and $f \in \mathcal{H}_{\boldsymbol{k}}$, [35] derived the following lower bound

$$\mathbb{E}_X \left( (f - \hat{f}_X)^2 \right) \ge \sum_{i=n+1}^\infty \alpha_i,$$

where $\alpha_i$ are the eigenvalues of $f$. Both of these bounds are equivalent asymptotically up to a constant for NTK (with bias) and the Laplace kernel.

### 3.4 NTK vs. exponential kernels in $\mathbb{R}^d$

While theoretical discussions of NTK largely assume the input data is normalized to lie on the sphere, such normalization is not the common practice in neural network applications. Instead, most often each feature is normalized separately by setting its mean to zero and variance to 1. Other normalizations are also common. It is therefore important to examine how NTK behaves outside of the hypersphere, compared to common kernels.

Below we derive the eigenfunctions of NTK for deep FC networks with ReLU activation with and without bias. We note that [5, 11] derived the eigenfunctions of NTK for two-layer FC networks with no bias. We will show that the same eigenfunctions are obtained with deep, bias-free networks, and that additional eigenfunctions appear when bias is added. We begin with a definition.

**Definition 1.** *A kernel $k$ is homogeneous of order $\alpha$ if $k(\mathbf{x}, \mathbf{z}) = \|\mathbf{x}\|^\alpha \|\mathbf{z}\|^\alpha k \left( \frac{\mathbf{x}^T \mathbf{z}}{\|\mathbf{x}\|\|\mathbf{z}\|} \right)$.*

**Theorem 4.** *(1) Bias-free $\boldsymbol{k}^{\mathrm{FC}_0(\mathrm{L})}$ is homogeneous of order 1. (2) With bias initialized at zero, let $\boldsymbol{k}^{\mathrm{Bias}(\mathrm{L})} = \boldsymbol{k}^{\mathrm{FC}_\beta(\mathrm{L})} - \boldsymbol{k}^{\mathrm{FC}_0(\mathrm{L})}$. Then, $\boldsymbol{k}^{\mathrm{Bias}(\mathrm{L})}$ is homogeneous of order 0.*

The two kernels $\boldsymbol{k}^{\mathrm{FC}_0(\mathrm{L})}$ and $\boldsymbol{k}^{\mathrm{FC}_\beta(\mathrm{L})}$ (but not $\boldsymbol{k}^{\mathrm{Bias}(\mathrm{L})}$) are unbounded. Therefore, their Mercer's representation (1) exists under measures that decay sufficiently fast as $\|\mathbf{x}\| \to \infty$. Examples include the uniform distribution on the $\|\mathbf{x}\| \le 1$ disk or the standard normal distribution. Such distributions have the virtue of being uniform on all concentric spheres. The following theorem determines the eigenfunctions for these kernels.

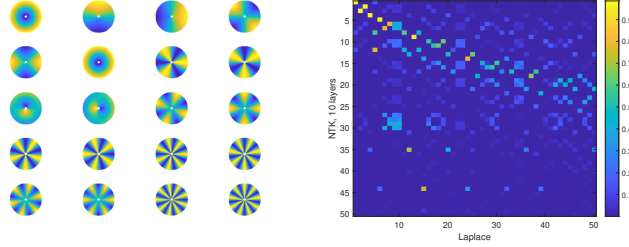

Figure 2: Left: plots of the eigenfunctions of NTK for a two layer FC network with bias on the unit disk, arranged in decreasing order of the eigenvalues. The radial shape of the eigenfunctions is evident. For two layers, the eigenvalues of $k^{\mathrm{FC}_0(2)}$ are zero for odd $k \geq 3$, while those of $k^{\mathrm{Bias}(2)}$ are zero for even $k \geq 2$. Therefore we see two "DC components" (top left and 2nd in 2nd row) and four $k = 1$ components (2nd and 3rd in 1st row and 1st and 2nd in third row). The rest of the frequencies are represented twice each. Right: Absolute correlation between the eigenfunctions of NTK and those of the Laplace kernel for data sampled uniformly on the unit disk. It can be seen that eigenfunctions of higher frequency for NTK correlate with eigenfunctions of higher frequency for the Laplace. However, relatively low order components for NTK contain higher frequency components of the Laplace.

**Theorem 5.** *Let $p(r)$ be a decaying density on $[0, \infty)$ such that $0 < \int_0^\infty p(r)r^2 dr < \infty$ and $\mathbf{x}, \mathbf{z} \in \mathbb{R}^d$.*

1. *Let $k_0(\mathbf{x}, \mathbf{z})$ be homogeneous of order 1 such that $k_0(\mathbf{x}, \mathbf{z}) = \|\mathbf{x}\| \, \|\mathbf{z}\| \, \hat{k}_0\left(\frac{\mathbf{x}^T \mathbf{z}}{\|\mathbf{x}\|\|\mathbf{z}\|}\right)$. Then its eigenfunctions with respect to $p(\|\mathbf{x}\|)$ are given by $\Psi_{k,j} = a\|\mathbf{x}\|Y_{k,j}\left(\frac{\mathbf{x}}{\|\mathbf{x}\|}\right)$ where $Y_{k,j}$ are the spherical harmonics in $\mathbb{S}^{d-1}$ and $a \in \mathbb{R}$.*

2. *Let $k(\mathbf{x}, \mathbf{z}) = k_0(\mathbf{x}, \mathbf{z}) + k_1(\mathbf{x}, \mathbf{z})$ so that $k_0$ as in 1 and $k_1$ is homogeneous of order 0. Then the eigenfunctions of $k$ are of the form $\Psi_{k,j} = (a \, \|\mathbf{x}\| + b) \, Y_{k,j}\left(\frac{\mathbf{x}}{\|\mathbf{x}\|}\right)$.*

The eigenfunctions of NTK in $\mathbb{R}^d$, therefore, are similar to those in $\mathbb{S}^{d-1}$; they are the spherical harmonics scaled radially in the bias free case, or linearly with the norm when bias is used. With bias, $k^{\mathrm{FC}_\beta(\mathrm{L})}$ has up to $2N(d, k)$ eigenfunctions for every frequency $k$. Compared to the eigenvalues in $\mathbb{S}^{d-1}$, the eigenvalues can change, depending on the radial density $p(r)$, but they maintain their overall asymptotic behavior.

In contrast to NTK, the Laplace kernel is shift invariant, and therefore its eigenfunctions are the Fourier transform. The two kernels hence cannot be compared merely by their eigenvalues. Figure 2 shows the eigenfunctions of NTK along with their correlation to the eigenfunctions of the Laplace kernel. While these differences are large, they seem to make only little difference in experiments, see Section 4 below. It is possible to produce a homogeneous version of the Laplace kernel as follows

$$k^{\mathrm{HLap}}(\mathbf{x}, \mathbf{z}) = \|\mathbf{x}\|\|\mathbf{z}\| \exp\left(-c\sqrt{1 - \frac{\mathbf{x}^T\mathbf{z}}{\|\mathbf{x}\|\|\mathbf{z}\|}}\right). \tag{5}$$

Following Thm. 5 the eigenfunctions of this kernel are the scaled spherical harmonics and, following Thm. 2, its eigenvalues decay at the rate of $k^{-d}$, much like the NTK.

## 4 Experiments

We compare the performance of NTK with Laplace, Gaussian, and $\gamma$-exponential kernels on both small and large scale real datasets. Our goal is to demonstrate: a) Results with the Laplace kernel are quite similar to those obtained by NTK, and b) The $\gamma$-exponential kernel can achieve slightly better results than NTK. Experimental details are provided in the supplementary material.

## 4.1 UCI dataSet

We compare methods using the same set of 90 small scale UCI datasets (with less than 5000 data points) as in [4]. The results are provided in Table 1 for the exponential kernels and their homogeneous versions, denoted by the "H-" prefix, as well as for NTK. For completeness, we also cite the results for Random forest (RF), the top classifier identified in [21], and neural networks from [4]. Further comparison of the accuracies obtained with NTK vs. the H-Laplace kernel on each of the 90 datasets is shown in Figure 3.

We report the same metrics as used in [4]: Friedman Ranking, Average Accuracy, P90/P95, and PMA. A superior classifier is expected to have lower Friedman rank and higher P90, P95, and PMA. Friedman Ranking [19] reports the average ranking of a given classifier compared to other classifiers. P90/P95 denotes the fraction of datasets on which a classifier achieves more than $90/95\%$ of the maximum achievable accuracy (i.e., maximum accuracy among all the classifiers [21]). PMA represents the percentage of maximum accuracy.

From Table 1, one can observe that the H-Laplace kernel results are the closest to NTK on all the metrics. In fact, as seen in Figure 3, these methods seem to achieve highly similar accuracies in each of the 90 datasets. Furthermore, the H-$\gamma$-exponential outperforms all the classifiers including NTK on all metrics. Moreover, the homogeneous versions slightly outperform the standard kernels. All these methods have hyperparameters that can be optimized. In [4], they search 105 hyperparameters for NTK. For a fair comparison, we search for the same number for the $\gamma$-exponential and fewer (70) for the Laplace kernels. We note finally that deeper networks yield NTK shapes that are more sharply peaked, corresponding to Laplace kernels with higher values of $c$. This is shown in Fig. 4 below.

| Classifier | F-Rank | Average Accuracy | P90 | P95 | PMA |
|---|---|---|---|---|---|
| H-$\gamma$-exp. | **26.26** | **82.25%$\pm$14.07%** | **92.22%** | **73.33%** | **96.07% $\pm$4.83%** |
| $\gamma$-exp. | 32.98 | 81.80%$\pm$14.21% | 85.56% | 73.33% | 95.49% $\pm$5.31% |
| H-Laplace | 29.60 | 81.74%$\pm$13.82% | 88.89% | 66.67% | 95.53% $\pm$4.84% |
| Laplace | 33.28 | 81.12%$\pm$14.16% | 86.67% | 65.56% | 94.88% $\pm$6.85% |
| H-Gaussian | 32.66 | 81.46% $\pm$ 14.83% | 84.44% | 67.77% | 94.95% $\pm$6.25% |
| Gaussian | 35.76 | 81.03% $\pm$ 15.09% | 85.56% | 72.22% | 94.56% $\pm$8.22% |
| NTK [4] | 28.34 | 81.95%$\pm$14.10% | 88.89% | 72.22% | 95.72% $\pm$5.17% |
| NN [4] | 38.06 | 81.02%$\pm$14.47% | 85.56% | 60.00% | 94.55% $\pm$5.89% |
| RF [4] | 33.51 | 81.56% $\pm$13.90% | 85.56% | 67.78% | 95.25% $\pm$5.30% |

Table 1: Performance on the UCI dataset. Lower F-Rank and higher P90, P95, PMA are better numbers.

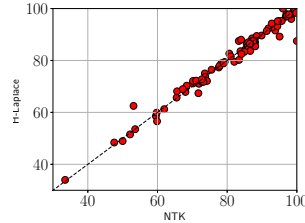

Figure 3: Performance comparisons between NTK and H-Laplace on the UCI dataset.

## 4.2 Large scale datasets

We leverage FALKON [42], an efficient approximate kernel method to conduct large scale regression and classification tasks following the setup of [42]. The results and datasets details are reported in Table 2. We searched for hyperparameters based on a small validation dataset for all the methods and used the standard train/test partition provided on the UCI repository. From Table 2, one can notice that NTK and H-Laplace perform similarly. For each dataset, either the $\gamma$-exponential or Gaussian kernels slightly outperforms these two kernels.

| | MillionSongs [10] | SUSY [40] | HIGGS [40] |
|---|---|---|---|
| #Training Data | $4.6 \times 10^5$ | $5 \times 10^6$ | $1.1 \times 10^7$ |
| #Features | 90 | 18 | 28 |
| Problem Type | Regression | Classification | Classification |
| Performance Metric | MSE | AUC | AUC |
| H-$\gamma$-exp. | **78.6417** | 87.686 | **82.281** |
| H-Laplace | 79.7941 | 87.670 | 81.995 |
| NTK | 79.9666 | 87.673 | 82.089 |
| H-Gaussian | 79.6255 | **87.689** | 81.967 |
| Neural Network [40] | - | 87.500 | 81.600 |
| Deep Neural Network [40] | - | **87.900**$^*$ | **88.500**$^*$ |

Table 2: Performance on the large scale datasets. We report MSE (lower is better) for the regression problem, and AUC (higher is better) for the classification problems.

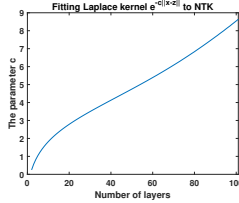

Figure 4: Fitting the Laplace kernel to NTK. The graph shows optimal width ($c$) of the Laplace kernel that is fitted to NTK with different number of layers.

## 4.3 Hierarchical convolutional kernels

Convolutional NTKs (CNTK) were shown to express the limit of convolutional neural networks when the number of channels tends to infinity, and recent empirical results showed that the two achieve similar accuracy on test data [2, 31]. CNTK is defined roughly by recursively applying NTK to image patches. For our final experiment we constructed alternative hierarchical kernels, in the spirit of [12, 33], by recursively applying exponential kernels in a manner similar to CNTK. The new kernel, denoted C-Exp, is applied first to pairs of $3 \times 3$ image patches, then to $3 \times 3$ patches of kernel values, and so forth. A detailed algorithm is provided in the supplementary material. We applied the kernel (using the homogeneous versions of the Laplace, Gaussian and $\gamma$-exponential kernels) to the Cifar-10 dataset and compared it to CNTK. Our experimental conditions and results for CNTK are identical to those of [2]. (Note that these do not include global average pooling.) Consistent with our previous experiments, Table 3 shows that these kernels are on par with the CNTK with small advantage to the $\gamma$-exponential kernel. This demonstrates that the four kernels maintain similar performance even after repeated application.

| Method | Accuracy (50k) | Accuracy(2k) |
|---|---|---|
| CNTK | 66.4% | 43.9% |
| C-Exp Laplace | 65.2% | 44.2% |
| C-Exp $\gamma$-exponential | **67.0%** | **45.2%** |
| C-Exp Gaussian | 66.8% | 45.0% |

Table 3: Classification accuracy for the CIFAR-10 dataset for our C-Exp hierarchical kernels, compared to CNTK. The two columns show results with training on the full dataset and on the first 2000 examples.

## 5 Conclusions

Our paper has considered the relationship between NTK and the classic Laplace kernel. Our main result is to show that for data normalized on the unit hypersphere, these two kernels have the same RKHS. Experiments show that the two kernels perform almost identically on a wide range of real-world applications. Coupled with prior results that show that kernel methods using NTK mimic the behavior of FC neural networks, our results suggest that much insight about neural networks can be obtained from analysis of the well-known Laplace kernel, which has a simple closed form.

Neural networks do offer great flexibility not easily translated to kernel methods. They can naturally be applied to large data sets, and researchers have developed many techniques for training them, such as dropout and batch normalization, that improve performance but do not directly translate to kernel methods. Furthermore, while we study feed-forward fully connected networks, analyzing more complex architectures, such as CNNs, GANs, autoencoders and recurrent networks, or the effect of other activation functions, remains a significant challenge for future work. Comparing the eigenfunctions of NTK with those of classical kernels under non-uniform distribution is yet a further challenge.

## Broader impact

This work explains the success of deep, fully connected networks through their similarity to exponential kernels. Such an analysis may allow for a better interpretability of deep network models.

## Acknowledgements

The authors thank the U.S.- Israel Binational Science Foundation, grant number 2018680, the National Science Foundation, grant no. IIS-1910132, the Quantifying Ensemble Diversity for Robust Machine Learning (QED for RML) program from DARPA and the Guaranteeing AI Robustness Against Deception (GARD) program from DARPA for their support of this project.

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
