[Supplementary Material]

# On the Similarity between the Laplace and Neural Tangent Kernels

## – Supplementary Material –

**Amnon Geifman**[1]    **Abhay Yadav**[2]    **Yoni Kasten**[1]

**Meirav Galun**[1]    **David Jacobs**[2]    **Ronen Basri**[1]

[1]Department of Computer Science, Weizmann Institute of Science, Rehovot, Israel
[2]Department of Computer Science, University of Maryland, College Park, MD

## A  Formulas for NTK

We begin by providing the recursive definition of NTK for fully connected (FC) networks with bias initialized at zero. The formulation includes a parameter $\beta$ that when set to zero the recursive formula coincides with the formula given in [1] for bias-free networks.

**The network model.**    We consider a $L$-hidden-layer fully-connected neural network (in total $L+1$ layers) with bias. Let $\mathbf{x} \in \mathbb{R}^d$ (and denote $d_0 = d$), we assume each layer $l \in [L]$ of hidden units includes $d_l$ units. The network model is expressed as

$$\mathbf{g}^{(0)}(\mathbf{x}) = \mathbf{x}$$
$$\mathbf{f}^{(l)}(\mathbf{x}) = W^{(l)}\mathbf{g}^{(l-1)}(\mathbf{x}) + \beta \mathbf{b}^{(l)} \in \mathbb{R}^{d_l}, \quad l = 1, \ldots L$$
$$\mathbf{g}^{(l)}(\mathbf{x}) = \sqrt{\frac{c_\sigma}{d_l}} \sigma\left(\mathbf{f}^{(l)}(\mathbf{x})\right) \in \mathbb{R}^{d_l}, \quad l = 1, \ldots L$$
$$f(\theta, \mathbf{x}) = f^{(L+1)}(\mathbf{x}) = W^{(L+1)} \cdot \mathbf{g}^{(L)}(\mathbf{x}) + \beta b^{(L+1)}$$

The network parameters $\theta$ include $W^{(L+1)}, W^{(L)}, ..., W^{(1)}$, where $W^{(l)} \in \mathbb{R}^{d_l \times d_{l-1}}$, $\mathbf{b}^{(l)} \in \mathbb{R}^{d_l \times 1}$, $W^{(L+1)} \in \mathbb{R}^{1 \times d_L}$, $b^{(L+1)} \in \mathbb{R}$, $\sigma$ is the activation function and $c_\sigma = 1/\left(\mathbb{E}_{z \sim \mathcal{N}(0,1)}[\sigma(z)^2]\right)$. The network parameters are initialized with $\mathcal{N}(0, I)$, except for the biases $\{\mathbf{b}^{(1)}, \ldots, \mathbf{b}^{(L)}, b^{(L+1)}\}$, which are initialized with zero.

**The recursive formula for NTK.**    The recursive formula in [9] assumes the bias is initialized with a normal distribution. Here we assume the bias is initialized at zero, yielding a sightly different formulation, which can be readily derived from [9]'s formulation.

Given $\mathbf{x}, \mathbf{z} \in \mathbb{R}^d$, we denote the NTK for this fully connected network with bias by $\mathbf{k}^{\mathrm{FC}_\beta(L+1)}(\mathbf{x}, \mathbf{z}) := \Theta^{(L)}(\mathbf{x}, \mathbf{z})$. The kernel $\Theta^{(L)}(\mathbf{x}, \mathbf{z})$ is defined using the following recursive definition. Let $h \in [L]$ then

$$\Theta^{(h)}(\mathbf{x}, \mathbf{z}) = \Theta^{(h-1)}(\mathbf{x}, \mathbf{z})\dot{\Sigma}^{(h)}(\mathbf{x}, \mathbf{z}) + \Sigma^{(h)}(\mathbf{x}, \mathbf{z}) + \beta^2, \tag{1}$$

where

$$\Sigma^{(0)}(\mathbf{x}, \mathbf{z}) = \mathbf{x}^T \mathbf{z}$$
$$\Theta^{(0)}(\mathbf{x}, \mathbf{z}) = \Sigma^{(0)}(\mathbf{x}, \mathbf{z}) + \beta^2.$$

and we define

$$\Sigma^{(h)}(\mathbf{x}, \mathbf{z}) = c_\sigma \mathbb{E}_{(u,v) \backsim N(0, \Lambda^{(h-1)})} \left( \sigma(u) \sigma(v) \right)$$

$$\dot{\Sigma}^{(h)}(\mathbf{x}, \mathbf{z}) = c_\sigma \mathbb{E}_{(u,v) \backsim N(0, \Lambda^{(h-1)})} \left( \dot{\sigma}(u) \dot{\sigma}(v) \right)$$

$$\Lambda^{(h-1)} = \begin{pmatrix} \Sigma^{(h-1)}(\mathbf{x}, \mathbf{x}) & \Sigma^{(h-1)}(\mathbf{x}, \mathbf{z}) \\ \Sigma^{(h-1)}(\mathbf{z}, \mathbf{x}) & \Sigma^{(h-1)}(\mathbf{z}, \mathbf{z}) \end{pmatrix}.$$

Now, let

$$\lambda^{(h-1)}(\mathbf{x}, \mathbf{z}) = \frac{\Sigma^{(h-1)}(\mathbf{x}, \mathbf{z})}{\sqrt{\Sigma^{(h-1)}(\mathbf{x}, \mathbf{x}) \Sigma^{(h-1)}(\mathbf{z}, \mathbf{z})}}. \tag{2}$$

By definition $|\lambda^{(h-1)}| \leq 1$, and for ReLU activation we have $c_\sigma = 2$ and

$$\Sigma^{(h)}(\mathbf{x}, \mathbf{z}) = c_\sigma \frac{\lambda^{(h-1)}(\pi - \arccos(\lambda^{(h-1)})) + \sqrt{1 - (\lambda^{(h-1)})^2}}{2\pi} \sqrt{\Sigma^{(h-1)}(\mathbf{x}, \mathbf{x}) \Sigma^{(h-1)}(\mathbf{z}, \mathbf{z})} \tag{3}$$

$$\dot{\Sigma}^{(h)}(\mathbf{x}, \mathbf{z}) = c_\sigma \frac{\pi - \arccos(\lambda^{(h-1)})}{2\pi}. \tag{4}$$

The parameter $\beta$ allows us to consider a fully-connected network either with ($\beta > 0$) or without bias ($\beta = 0$). When $\beta = 0$, the recursive formulation is the same as existing derivations, e.g., [9]. Finally, the normalized NTK of a FC network with $L + 1$ layers, without bias, is given by $\frac{1}{L+1} \boldsymbol{k}^{\mathrm{FC}_0(\mathrm{L}+1)}(\mathbf{x}_i, \mathbf{x}_j)$.

**NTK for a two-layer FC network on $\mathbb{S}^{d-1}$.** Using the recursive formulation above, for points on the hypersphere $\mathbb{S}^{d-1}$ NTK for a two-layer FC network with bias initialized at 0, is as follows. Let $u = \mathbf{x}^T \mathbf{z}$, with $\mathbf{x}, \mathbf{z} \in \mathbb{S}^{d-1}$. Then,

$$\begin{aligned}
\boldsymbol{k}^{\mathrm{FC}_\beta(2)}(\mathbf{x}, \mathbf{z}) &= \Theta^{(1)}(\mathbf{x}, \mathbf{z}) \\
&= \Theta^{(0)}(\mathbf{x}, \mathbf{z}) \dot{\Sigma}^{(1)}(\mathbf{x}, \mathbf{z}) + \Sigma^{(1)}(\mathbf{x}, \mathbf{z}) + \beta^2 \\
&= (u + \beta^2) \frac{\pi - \arccos(u)}{\pi} + \frac{u(\pi - \arccos(u)) + \sqrt{1 - u^2}}{\pi} + \beta^2.
\end{aligned}$$

Rearranging, we get

$$\boldsymbol{k}^{\mathrm{FC}_\beta(2)}(\mathbf{x}, \mathbf{z}) = \boldsymbol{k}^{\mathrm{FC}_\beta(2)}(u) = \frac{1}{\pi} \left( (2u + \beta^2)(\pi - \arccos(u)) + \sqrt{1 - u^2} \right) + \beta^2. \tag{5}$$

# B   NTK on $\mathbb{S}^{d-1}$

This section provides a characterization of NTK on the hypersphere $\mathbb{S}^{d-1}$ under the uniform measure. The recursive formulas of the kernels are given in Appendix A.

**Lemma 1.** *Let $\boldsymbol{k}^{\mathrm{FC}_\beta(\mathrm{L})}(\mathbf{x}, \mathbf{z})$, $\mathbf{x}, \mathbf{z} \in \mathbb{S}^{d-1}$, denote the NTK kernels for FC networks with $L \geq 2$ layers, possibly with bias initialized with zero. This kernel is zonal, i.e., $\boldsymbol{k}^{\mathrm{FC}_\beta(\mathrm{L})}(\mathbf{x}, \mathbf{z}) = \boldsymbol{k}^{\mathrm{FC}_\beta(\mathrm{L})}(\mathbf{x}^T \mathbf{z})$.*

*Proof.* See Appendix D.

$\square$

To prove the next theorem, we recall several results on the the arithmetics of RKHS, following [8, 15].

## B.1   RKHS for sums and products of kernels.

Let $\boldsymbol{k}_1, \boldsymbol{k}_2 : \mathcal{X} \times \mathcal{X} \to \mathbb{R}$ be kernels with RKHS $\mathcal{H}_{\boldsymbol{k}_1}$ and $\mathcal{H}_{\boldsymbol{k}_2}$, respectively. Then,

    1. **Aronszajn's kernel sum theorem.** The RKHS for $\boldsymbol{k} = \boldsymbol{k}_1 + \boldsymbol{k}_2$ is given by $\mathcal{H}_{\boldsymbol{k}_1 + \boldsymbol{k}_2} = \{f_1 + f_2 \mid f_1 \in \mathcal{H}_{\boldsymbol{k}_1}, \ f_2 \in \mathcal{H}_{\boldsymbol{k}_2}\}$

2. This yields the **kernel sum inclusion.** $\mathcal{H}_{\boldsymbol{k}_1}, \mathcal{H}_{\boldsymbol{k}_2} \subseteq \mathcal{H}_{\boldsymbol{k}_1 + \boldsymbol{k}_2}$

3. **Norm addition inequality.** $\|f_1 + f_2\|_{\mathcal{H}_{\boldsymbol{k}_1 + \boldsymbol{k}_2}} \leq \|f_1\|_{\mathcal{H}_{\boldsymbol{k}_1}} + \|f_2\|_{\mathcal{H}_{\boldsymbol{k}_2}}$

4. **Norm product inequality.** $\|f_1 \cdot f_2\|_{\mathcal{H}_{\boldsymbol{k}_1 \cdot \boldsymbol{k}_2}} \leq \|f_1\|_{\mathcal{H}_{\boldsymbol{k}_1}} \cdot \|f_2\|_{\mathcal{H}_{\boldsymbol{k}_2}}$

5. **Aronszajn's inclusion theorem.** $\mathcal{H}_{\boldsymbol{k}_1} \subseteq \mathcal{H}_{\boldsymbol{k}_2}$ if and only if $\exists s > 0$, such that $\boldsymbol{k}_1 \ll s^2 \boldsymbol{k}_2$, where the latter notation means that $s^2 \boldsymbol{k}_2 - \boldsymbol{k}_1$ is a positive definite kernel over $\mathcal{X}$.

## B.2 The decay rate of the eigenvalues of NTK

**Theorem 1.** *Let* $\mathbf{x}, \mathbf{z} \in \mathbb{S}^{d-1}$. *With bias initialized at zero and* $\beta > 0$:

1. $\boldsymbol{k}^{\mathrm{FC}_\beta(\mathrm{L})}$ *can be decomposed according to*

$$\boldsymbol{k}^{\mathrm{FC}_\beta(\mathrm{L})}(\mathbf{x}, \mathbf{z}) = \sum_{k=0}^{\infty} \lambda_k \sum_{j=1}^{N(d,k)} Y_{k,j}(\mathbf{x}) Y_{k,j}(\mathbf{z}), \tag{6}$$

*with* $\lambda_k > 0$ *for all* $k \geq 0$ *and into* $Y_{k,j}$ *are the spherical harmonics of* $\mathbb{S}^{d-1}$, *and*

2. $\exists k_0$ *and constants* $C_1, C_2, C_3 > 0$ *that depend on the dimension* $d$ *such that* $\forall k > k_0$

   (a) $C_1 k^{-d} \leq \lambda_k \leq C_2 k^{-d}$ *if* $L = 2$, *and*
   (b) $C_3 k^{-d} \leq \lambda_k$ *if* $L \geq 3$.

We split the theorem into the next two lemmas. The first lemma handles NTK of two-layer FC networks with bias, and the second lemma handles NTK for deep networks.

**Lemma 2.** *Let* $\mathbf{x}, \mathbf{z} \in \mathbb{S}^{d-1}$ *and* $\boldsymbol{k}^{\mathrm{FC}_\beta(2)}(\mathbf{x}^T \mathbf{z})$ *as defined in* (5) *with* $\beta > 0$. *Then,* $\boldsymbol{k}^{\mathrm{FC}_\beta(2)}$ *decomposes according to* (6) *where* $\lambda_k > 0$ *for all* $k \geq 0$ *and* $\exists k_0$ *such that* $\forall k \geq k_0$

$$C_1 k^{-d} \leq \lambda_k \leq C_2 k^{-d},$$

*where* $C_1, C_2 > 0$ *are constants that depend on the dimension* $d$.

*Proof.* To prove the lemma we leverage the results of [3, 5]. First, under the assumption of the uniform measure on $\mathbb{S}^{d-1}$, we can apply Mercer decomposition to $\boldsymbol{k}^{\mathrm{FC}_\beta(2)}(\mathbf{x}, \mathbf{z})$, where the eigenfunctions are the spherical harmonics. This is due to the observation that $\boldsymbol{k}^{\mathrm{FC}_\beta(2)}(\mathbf{x}, \mathbf{z})$ is positive and zonal in $\mathbb{S}^{d-1}$. It is zonal by Lemma 1 and positive, since $\boldsymbol{k}^{\mathrm{FC}_\beta(2)}$ can be decomposed as

$$\begin{aligned}
\boldsymbol{k}^{\mathrm{FC}_\beta(2)}(u) &= \frac{1}{\pi} \left( (2u + \beta^2)(\pi - \arccos(u)) + \sqrt{1 - u^2} \right) + \beta^2 \\
&= \frac{1}{\pi} \left( 2u(\pi - \arccos(u)) + \sqrt{1 - u^2} \right) + \frac{1}{\pi} \beta^2 (\pi - \arccos(u)) + \beta^2 \\
&:= \kappa(\mathbf{x}^T \mathbf{z}) + \beta^2 \kappa_0(\mathbf{x}^T \mathbf{z}) + \beta^2,
\end{aligned}$$

where $\kappa(\mathbf{x}^T \mathbf{z})$ is the NTK for a bias-free, two-layer network introduced in [5] and $\kappa_0(\mathbf{x}^T \mathbf{z})$ is known to be the zero-order arc-cosine kernel [6]. By kernel arithmetic, this yields another kernel and this means that $\boldsymbol{k}^{\mathrm{FC}_\beta(2)}$ is a positive kernel.

Furthermore, according to Proposition 5 in [5]

$$\kappa(\mathbf{x}^T \mathbf{z}) = \sum_{k=0}^{\infty} \mu_k \sum_{j=1}^{N(d,k)} Y_{k,j}(\mathbf{x}) Y_{k,j}(\mathbf{z}),$$

where $Y_{k,j}, j = 1, \ldots, N(d, k)$ are spherical harmonics of degree $k$, and the eigenvalues $\mu_k$ satisfy $\mu_0, \mu_1 > 0$, $\mu_k = 0$ if $k = 2j + 1$ with $j \geq 1$ and otherwise, $\mu_k > 0$ and $\mu_k \sim C(d)k^{-d}$ as $k \to \infty$, with $C(d)$ a constant depending only on $d$. Next, following Lemma 17 in [5] the eigenvalues of $\kappa_0(\mathbf{x}^T \mathbf{z})$, denoted $\eta_k$ satisfy $\eta_0, \eta_1 > 0$, $\eta_k > 0$ if $k = 2j + 1$, with $j \geq 1$ and behave asymptotically as $C_0(d)k^{-d}$. Consequently, $\boldsymbol{k}^{\mathrm{FC}_\beta(2)} = \kappa + \beta^2 \kappa_0 + \beta^2$, and since both $\kappa$ and $\kappa_0$ have the spherical

harmonics as their eigenfunctions, their eigenvalues are given by $\lambda_k = \mu_k + \beta^2 \eta_k > 0$ for $k > 0$ and $\lambda_0 = \mu_0 + \beta^2 \eta_0 + \beta^2 > 0$, and asymptotically $\lambda_k \sim \tilde{C}(d)k^{-d}$, where $\tilde{C}(d) = C(d) + \beta^2 C_0(d)$.

To conclude, this implies that $\exists k_0, C_1(d) > 0$ and $C_2(d) > 0$, such that for all $k \geq k_0$ it holds that

$$C_1 k^{-d} \leq \lambda_k \leq C_2 k^{-d}$$

and also, unless $\beta = 0$, for all $k \geq 0$

$$\lambda_k > 0.$$

$\square$

Next, we prove the second part of Theorem 1 that relates to deep FC networks with bias, $\boldsymbol{k}^{\mathrm{FC}_\beta(\mathrm{L})}$, i.e. we prove the following lemma.

**Lemma 3.** *Let* $\mathbf{x}, \mathbf{z} \in \mathbb{S}^{d-1}$ *and* $\boldsymbol{k}^{\mathrm{FC}_\beta(\mathrm{L})}(\mathbf{x}^T \mathbf{z})$ *as defined in Appendix A. Then*

1. $\boldsymbol{k}^{\mathrm{FC}_\beta(\mathrm{L})}$ *decomposes according to* (6) *with* $\lambda_k > 0$ *for all* $k \geq 0$

2. $\exists k_0$ *such that* $\forall k > k_0$ *it holds that* $C_3 k^{-d} \leq \lambda_k$ *in which* $C_3 > 0$ *depends on the dimension* $d$

3. $\mathcal{H}^{\mathrm{FC}_\beta(\mathrm{L}-1)} \subseteq \mathcal{H}^{\mathrm{FC}_\beta(\mathrm{L})}$

*Proof.* Following Lemma 1, it holds that $\boldsymbol{k}^{\mathrm{FC}_\beta(\mathrm{L})}$ is zonal, and therefore can be decomposed according to (6). In order to prove the lemma we look at the recursive formulation of the NTK kernel, i.e.,

$$\boldsymbol{k}^{\mathrm{FC}_\beta(\mathrm{l+1})} = \boldsymbol{k}^{\mathrm{FC}_\beta(\mathrm{l})} \dot{\Sigma}^{(l)} + \Sigma^{(l)} + \beta^2. \tag{7}$$

Now, following Lemma 17 in [5] all of the eigenvalues of $\dot{\Sigma}^{(l)}$ are positive, including $\lambda_0 > 0$. This implies that the constant function $g(\mathbf{x}) \equiv 1 \in \mathcal{H}_{\dot{\Sigma}^{(l)}}$.

Now, we use the norm multiplicity inequality in Sec. B.1 and show that $\mathcal{H}_{\boldsymbol{k}^{\mathrm{FC}_\beta(\mathrm{l})}} \subseteq \mathcal{H}_{\boldsymbol{k}^{\mathrm{FC}_\beta(\mathrm{l})} \cdot \dot{\Sigma}^{(l)}}$. Let $f \in \mathcal{H}_{\boldsymbol{k}^{\mathrm{FC}_\beta(\mathrm{l})}}$, i.e., $\|f\|_{\mathcal{H}_{\boldsymbol{k}^{\mathrm{FC}_\beta(\mathrm{l})}}} < \infty$. We showed that $1 \in \mathcal{H}_{\dot{\Sigma}^{(l)}}$. Therefore, $\|f \cdot 1\|_{\mathcal{H}_{\boldsymbol{k}^{\mathrm{FC}_\beta(\mathrm{l})} \cdot \dot{\Sigma}^{(l)}}} \leq \|f\|_{\mathcal{H}_{\boldsymbol{k}^{\mathrm{FC}_\beta(\mathrm{l})}}} \|1\|_{\mathcal{H}_{\dot{\Sigma}^{(l)}}} < \infty$, implying that $f \in \mathcal{H}_{\boldsymbol{k}^{\mathrm{FC}_\beta(\mathrm{l})} \cdot \dot{\Sigma}^{(l)}}$.

Finally, according to the kernel sum inclusion in Sec. B.1, relying on the recursive formulation (7) we have $\mathcal{H}_{\boldsymbol{k}^{\mathrm{FC}_\beta(\mathrm{l})}} \subseteq \mathcal{H}_{\boldsymbol{k}^{\mathrm{FC}_\beta(\mathrm{l})} \cdot \dot{\Sigma}^{(l)}} \subseteq \mathcal{H}_{\boldsymbol{k}^{\mathrm{FC}_\beta(\mathrm{l+1})}}$. Therefore,

$$\mathcal{H}^{\mathrm{FC}_\beta(2)} \subseteq \ldots \subseteq \mathcal{H}^{\mathrm{FC}_\beta(\mathrm{L}-1)} \subseteq \mathcal{H}^{\mathrm{FC}_\beta(\mathrm{L})}. \tag{8}$$

This completes the proof, by using Aronszan's inclusion theorem as follows. Since $H^{k^{FC(2)}} \subseteq H^{k^{FC(L)}}$, then by Aronszajn's inclusion theorem $\exists s > 0$ such that $\boldsymbol{k}^{\mathrm{FC}_\beta(2)} << s^2 \boldsymbol{k}^{\mathrm{FC}_\beta(\mathrm{L})}$. Since the kernels are zonal on the sphere (with uniform distribution of the data) their corresponding RKHS share the same eigenfunctions, namely the spherical harmonics.

Therefore, for all $k \geq 0$ it holds

$$s^2 \lambda_k^{\boldsymbol{k}^{\mathrm{FC}_\beta(\mathrm{L})}} \geq \lambda_k^{\boldsymbol{k}^{\mathrm{FC}_\beta(2)}} > 0$$

and for $k \to \infty$ it holds that

$$s^2 \lambda_k^{\boldsymbol{k}^{\mathrm{FC}_\beta(\mathrm{L})}} \geq \lambda_k^{\boldsymbol{k}^{\mathrm{FC}_\beta(2)}} \geq \frac{C_1}{k^d}$$

completing the proof.

$\square$

# C  Laplace Kernel in $\mathbb{S}^{d-1}$

The Laplace kernel $\boldsymbol{k}(\mathbf{x}, \mathbf{y}) = e^{-\bar{c}\|\mathbf{x}-\mathbf{y}\|}$ restricted to the sphere $\mathbb{S}^{d-1}$ is defined as

$$K(\mathbf{x}, \mathbf{y}) = \boldsymbol{k}(\mathbf{x}^T\mathbf{y}) = e^{-c\sqrt{1-x^Ty}} \tag{9}$$

where $c > 0$ is a tuning parameter. We next prove an asymptotic bound on its eigenvalues.

**Theorem 2.** *Let* $\mathbf{x}, \mathbf{y} \in \mathbb{S}^{d-1}$ *and* $\boldsymbol{k}(\mathbf{x}^T\mathbf{y}) = e^{-c\sqrt{1-\mathbf{x}^T\mathbf{y}}}$ *be the Laplace kernel, restricted to* $\mathbb{S}^{d-1}$. *Then* $\boldsymbol{k}$ *can be decomposed as in* (6) *with the eigenvalues* $\lambda_k$ *satisfying* $\lambda_k > 0$ *for all* $k \geq 0$ *and* $\exists k_0$ *such that* $\forall k > k_0$ *it holds that:*
$$B_1 k^{-d} \leq \lambda_k \leq B_2 k^{-d}$$
*where* $B_1, B_2 > 0$ *are constants that depend on the dimension* $d$ *and the parameter* $c$.

Our proof relies on several supporting lemmas.

**Lemma 4.** *([17] Thm 1.14 page 6) For all* $\alpha > 0$ *it holds that*

$$\int_{\mathbb{R}^d} e^{-2\pi\|\mathbf{x}\|\alpha} e^{-2\pi i \mathbf{t}\cdot\mathbf{x}} d\mathbf{x} = c_d \frac{\alpha}{(\alpha^2 + \|\mathbf{t}\|^2)^{(d+1)/2}}, \tag{10}$$

*where* $c_d = \Gamma(\frac{d+1}{2})/(\pi^{(d+1)/2})$

**Lemma 5.** *Let* $f(\mathbf{x}) = e^{-c\|\mathbf{x}\|}$ *with* $\mathbf{x} \in \mathbb{R}^d$. *Then, its Fourier transform* $\Phi(\mathbf{w})$ *with* $\mathbf{w} \in \mathbb{R}^d$ *is* $\Phi(\mathbf{w}) = \Phi(\|\mathbf{w}\|) = C(1 + \|\mathbf{w}\|^2/c^2)^{-(d+1)/2}$ *for some constant* $C > 0$.

*Proof.* To calculate the Fourier transform we need to calculate the following integral

$$\Phi(\mathbf{w}) = \frac{1}{(2\pi)^d} \int_{\mathbb{R}^d} e^{-c\|\mathbf{x}\|} e^{-i\mathbf{x}\cdot\mathbf{w}} d\mathbf{x}.$$

According to the Lemma 4, plugging $\alpha = \frac{c}{2\pi}$ and $\mathbf{t} = \frac{\mathbf{w}}{2\pi}$ into (10) yields

$$\Phi(\mathbf{w}) = c_d \frac{c}{(c^2 + \|\mathbf{w}\|^2)^{(d+1)/2}} = \frac{c_d}{c^{(d+1)}} \frac{1}{\left(1 + \frac{\|\mathbf{w}\|^2}{c^2}\right)^{(d+1)/2}} = C\left(1 + \frac{\|\mathbf{w}\|^2}{c^2}\right)^{-(d+1)/2}$$

with $C = \frac{c_d}{c^{(d+1)}} > 0$.

$\square$

**Lemma 6.** *([11] Thm. 4.1) Let* $f(\mathbf{x})$ *be defined as* $f(\|\mathbf{x}\|)$ *for all* $\mathbf{x} \in \mathbb{R}^d$, *and let* $\Phi(\mathbf{w}) = \Phi(\|\mathbf{w}\|)$ *denote its Fourier Transform in* $\mathbb{R}^d$. *Then, its corresponding kernel on* $\mathbb{S}^{d-1}$ *is defined as the restriction* $\boldsymbol{k}(\mathbf{x}^T\mathbf{y}) = f(\|\mathbf{x} - \mathbf{y}\|)$ *with* $\mathbf{x}, \mathbf{y} \in \mathbb{S}^{d-1}$. *By Mercer's Theorem the spherical harmonic expansion of* $\boldsymbol{k}(\mathbf{x}^T\mathbf{y})$ *is of the form*

$$\boldsymbol{k}(\mathbf{x}^T\mathbf{y}) = \sum_{k=0}^{\infty} \lambda_k \sum_{j=1}^{N(d,k)} Y_{k,j}(\mathbf{x}) Y_{k,j}(\mathbf{y}).$$

*Then, the eigenvalues in the spherical harmonic expansion* $\lambda_k$ *are related to the Fourier coefficients of* $f$, $\Phi(t)$, *as follows*

$$\lambda_k = \int_o^{\infty} t\Phi(t) J_{k+\frac{d-2}{2}}^2(t) dt, \tag{11}$$

*where* $J_v(t)$ *is the usual Bessel function of the first kind of order* $v$.

Having, these supporting Lemmas, we can now prove **Theorem 2**.

*Proof.* First, $\boldsymbol{k}(\cdot, \cdot)$ is a positive zonal kernel and hence can be written as

$$\boldsymbol{k}(\mathbf{x}^T\mathbf{y}) = \sum_{k=0}^{\infty} \lambda_k \sum_{j=1}^{N(d,k)} Y_{k,j}(\mathbf{x}) Y_{k,j}(\mathbf{y}).$$

Next, to derive the bounds we plug the Fourier coefficients, $\Phi(\omega)$, computed in Lemma 5, into the expression for the harmonic coefficients, $\lambda_k$ (11), obtaining

$$\lambda_k = C \int_0^\infty \frac{t}{\left(1 + \frac{t^2}{c^2}\right)^{\frac{d+1}{2}}} J^2_{k+\frac{d-2}{2}}(t)dt.$$

Applying a change of variables $t = cx$ we get

$$\lambda_k = c^2 C \int_0^\infty \frac{x}{(1 + x^2)^{\frac{d+1}{2}}} J^2_{k+\frac{d-2}{2}}(cx)dx. \tag{12}$$

We next bound this integral from both above and below. To get an upper bound we observe that for $x \in [0, \infty)$ $x^2 < 1 + x^2$, implying that $x(1 + x^2)^{-(d+1)/2} < x^{-d}$, and consequently

$$\lambda_k < c^2 C \int_0^\infty x^{-d} J^2_{k+\frac{d-2}{2}}(cx)dx := c^2 C A(k, d, c).$$

The above integral $A(k, d, c)$ was computed in [18] (Sec. 13.41 page 402 with $a := c$, $\lambda := d$, and $\mu = \nu := k + (d - 2)/2$) which gives

$$A(k, d, c) = \int_0^\infty x^{-d} J^2_{k+\frac{d-2}{2}}(cx)dx = \frac{(\frac{c}{2})^{d-1}\Gamma(d)\Gamma(k - \frac{1}{2})}{2\Gamma^2(\frac{d+1}{2})\Gamma(k + d - \frac{1}{2})}. \tag{13}$$

Using Stirling's formula $\Gamma(x) = \sqrt{2\pi}x^{x-1/2}e^{-x}(1 + O(x^{-1}))$ as $x \to \infty$. Consequently, for sufficiently large $k >> d$

$$\lambda_k < c^2 C A(k, d, c) = c^2 C \frac{(\frac{c}{2})^{d-1}\Gamma(d)\Gamma(k - \frac{1}{2})}{2\Gamma^2(\frac{d+1}{2})\Gamma(k + d - \frac{1}{2})}$$

$$\sim c^2 C \frac{(\frac{c}{2})^{d-1}\Gamma(d)}{2\Gamma^2(\frac{d+1}{2})} \cdot \frac{(k - \frac{1}{2})^{k-1}e^{-k+\frac{1}{2}}}{(k + d - \frac{1}{2})^{k+d-1}e^{-k-d+\frac{1}{2}}}(1 + O(k^{-1}))$$

$$= B_2 k^{-d}, \tag{14}$$

where $B_2$ depends on $c$, $C$ and the dimension $d$.

We use again the relation (12) to derive a lower bound for $\lambda_k$. First, note that since $t, 1 + t^2, J^2_\nu(t)$ are all non-negative for $t \in [0, \infty)$ and therefore

$$\lambda_k \geq c^2 C \int_1^\infty \frac{x}{(1 + x^2)^{\frac{d+1}{2}}} J^2_{k+\frac{d-2}{2}}(cx)dx \geq c^2 C \int_1^\infty \frac{1}{2^{\frac{d+1}{2}}x^d} J^2_{k+\frac{d-2}{2}}(cx)dx$$

$$= \frac{Cc^2}{2^{\frac{d+1}{2}}} \left( \int_0^\infty x^{-d} J^2_{k+\frac{d-2}{2}}(cx)dx - \int_0^1 x^{-d} J^2_{k+\frac{d-2}{2}}(cx)dx \right)$$

$$= \frac{Cc^2}{2^{\frac{d+1}{2}}} \int_0^\infty x^{-d} J^2_{k+\frac{d-2}{2}}(cx)dx \left( 1 - \frac{\int_0^1 x^{-d} J^2_{k+\frac{d-2}{2}}(cx)dx}{\int_0^\infty x^{-d} J^2_{k+\frac{d-2}{2}}(cx)dx} \right)$$

$$= \frac{Cc^2}{2^{\frac{d+1}{2}}} A(k, d, c) \left( 1 - \frac{B(k, d, c)}{A(k, d, c)} \right),$$

where $B(k, d, c) := \int_0^1 x^{-d} J^2_{k+\frac{d-2}{2}}(cx)dx$. The first integral, $A(k, d, c)$, was shown in (14) to converge asymptotically to $B_2 k^{-d}$. To bound the second integral, $B(k, d, c)$, we use an inequality from [18] (Section 3.31, page 49), which states that for $v, t \in \mathbb{R}$, $v > -\frac{1}{2}$,

$$|J_v(t)| \leq \frac{2^{-v}t^v}{\Gamma(v + 1)}.$$

This gives an upper bound for $B(k, d, c)$

$$B(k, d, c) = \int_0^1 x^{-d} J^2_{k+\frac{d-2}{2}}(cx)dx \leq \int_0^1 x^{-d} \frac{2^{-2(k+\frac{d-2}{2})}(cx)^{2(k+\frac{d-2}{2})}}{\Gamma^2(k + \frac{d}{2})}dx \leq \frac{(\frac{c}{2})^{2(k+\frac{d-2}{2})}}{\Gamma^2(k + \frac{d}{2})}.$$

Applying Stirling's formula we obtain $B(k,d,c) \le O\left(\frac{(\frac{ce}{2})^{2(k+\frac{d}{2})}(k+d)}{(k+\frac{d}{2})^{2(k+\frac{d}{2})}}\right)$, which implies that as $k$ grows, $\frac{B(k,d,c)}{A(k,d,c)} \to 0$. Therefore, asymptotically for large $k$

$$\lambda_k \ge \frac{Cc^2}{2^{\frac{d+1}{2}}} A(k,d,c)\left(1 - \frac{B(k,d,c)}{A(k,d,c)}\right) \ge \frac{Cc^2}{2^{\frac{d+1}{2}}} A(k,d,c),$$

from which we conclude that $\lambda_k > B_1 k^{-d}$, where the constant $B_1$ depends on $c$, $C$, and $d$. We have therefore shown that there exists $k_0$ such that $\forall k > k_0$

$$B_1 k^{-d} \le \lambda_k \le B_2 k^{-d}.$$

Finally, to show that $\lambda_k > 0$ for all $k \ge 0$ we use again (11) in Lemma 6 which states that

$$\lambda_k = \int_0^\infty t\Phi(t)J^2_{k+\frac{d-2}{2}}(t)dt.$$

Note that in the interval $(0,\infty)$ it holds that $t > 0$ and $\Phi(t) > 0$ due to Lemma 5. Therefore $\lambda_k = 0$ implies that $J^2_{k+\frac{d-2}{2}}(t)$ is identically 0 on $(0,\infty)$, contradicting the properties of the Bessel function of the first kind. Hence, $\lambda_k > 0$ for all $k$. $\qquad\square$

## C.1 Proof of main theorem

**Theorem 3.** *Let $\mathcal{H}^{\mathrm{Lap}}$ denote the RKHS for the Laplace kernel restricted to $\mathbb{S}^{d-1}$, and let $\mathcal{H}^{\mathrm{FC}_\beta(\mathrm{L})}$ denote the NTK corresponding to a FC network with $L$ layers with bias, restricted to $\mathbb{S}^{d-1}$, then $\mathcal{H}^{\mathrm{Lap}} = \mathcal{H}^{\mathrm{FC}_\beta(2)} \subseteq \mathcal{H}^{\mathrm{FC}_\beta(\mathrm{L})}$.*

*Proof.* Let $\lambda_k^{\mathrm{Lap}}$, $\lambda_k^{\mathrm{FC}_\beta(2)}$, and $\lambda_k^{\mathrm{FC}_\beta(\mathrm{L})}$ denote the eigenvalues of the three kernel, $k^{\mathrm{Lap}}$, $k^{\mathrm{FC}_\beta(2)}$, and $k^{\mathrm{FC}_\beta(\mathrm{L})}$ in their Mercer's decomposition, i.e.,

$$k(\mathbf{x}^T\mathbf{z}) = \sum_{k=0}^\infty \lambda_k \sum_{j=1}^{N(d,k)} Y_{k,j}(\mathbf{x})Y_{k,j}(\mathbf{z}).$$

Denote by $k_0$ the smallest $k$ for which Theorems 1 and 2 hold simultaneously. We first show that $\mathcal{H}^{\mathrm{Lap}} \subseteq \mathcal{H}^{\mathrm{FC}_\beta(2)}$. Let $f(\mathbf{x}) \in \mathcal{H}^{\mathrm{Lap}}$, and let $f(\mathbf{x}) = \sum_{k=0}^\infty \sum_{j=0}^{N(d,k)} \alpha_{k,j} Y_{k,j}(\mathbf{x})$ denote its spherical harmonic decomposition. Then $\|f\|_{\mathcal{H}^{\mathrm{Lap}}} < \infty$ implies, due to Theorem 2, that

$$\sum_{k=k_0}^\infty \sum_{j=0}^{N(d,k)} \frac{1}{B_2} k^d \alpha_{k,j}^2 \le \sum_{k=k_0}^\infty \sum_{j=0}^{N(d,k)} \frac{\alpha_{k,j}^2}{\lambda_k^{\mathrm{Lap}}} < \infty.$$

Combining this with Theorem 1, and recalling that $\lambda_k^{\mathrm{FC}_\beta(2)} > 0$ for all $k \ge 0$), we have

$$\sum_{k=k_0}^\infty \sum_{j=0}^{N(d,k)} \frac{\alpha_{k,j}^2}{\lambda_k^{\mathrm{FC}_\beta(2)}} \le \sum_{k=k_0}^\infty \sum_{j=0}^{N(d,k)} \frac{1}{C_1} k^d \alpha_{k,j}^2 = \frac{B_2}{C_1} \sum_{k=k_0}^\infty \sum_{j=0}^{N(d,k)} \frac{1}{B_2} k^d \alpha_{k,j}^2 < \infty,$$

implying that $\|f\|_{\mathcal{H}^{\mathrm{FC}_\beta(2)}}^2 < \infty$, and so $\mathcal{H}^{\mathrm{Lap}} \subseteq \mathcal{H}^{\mathrm{FC}_\beta(2)}$. Similar arguments can be used to show that $\mathcal{H}^{\mathrm{FC}_\beta(2)} \subseteq \mathcal{H}^{\mathrm{Lap}}$, proving that $\mathcal{H}^{\mathrm{FC}_\beta(2)} = \mathcal{H}^{\mathrm{Lap}}$. Finally, following the inclusion relation (8) the theorem is proved. $\qquad\square$

# D  NTK in $\mathbb{R}^d$

In this section we denote $r_x = \|\mathbf{x}\|$, $r_z = \|\mathbf{z}\|$ and by $\hat{\mathbf{x}} = \mathbf{x}/r_x$, $\hat{\mathbf{z}} = \mathbf{z}/r_z$. We first prove Theorem 4 and as a consequence Lemma 7 is proved.

**Theorem 4.** *Let $k^{\mathrm{FC}_0(\mathrm{L})}(\mathbf{x},\mathbf{z})$, $k^{\mathrm{FC}_\beta(\mathrm{L})}(\mathbf{x},\mathbf{z})$, $\mathbf{x},\mathbf{z} \in \mathbb{R}^d$, denote the NTK kernel with $L$ layers without bias and with bias initialized at zero, respectively. It holds that (1) Bias-free $k^{\mathrm{FC}_0(\mathrm{L})}$ is homogeneous of order 1. (2) Let $k^{\mathrm{Bias}(\mathrm{L})} = k^{\mathrm{FC}_\beta(\mathrm{L})} - k^{\mathrm{FC}_0(\mathrm{L})}$. Then, $k^{\mathrm{Bias}(\mathrm{L})}$ is homogeneous of order 0.*

**Lemma 7.** *Let $k^{\mathrm{FC}_\beta(\mathrm{L})}(\mathbf{x}, \mathbf{z})$, $\mathbf{x}, \mathbf{z} \in \mathbb{S}^{d-1}$, denote the NTK kernels for FC networks with $L \geq 2$ layers, possibly with bias initialized with zero. This kernel is zonal, i.e., $k^{\mathrm{FC}_\beta(\mathrm{L})}(\mathbf{x}, \mathbf{z}) = k^{\mathrm{FC}_\beta(\mathrm{L})}(\mathbf{x}^T \mathbf{z})$.*

To that end, we first prove the following supporting Lemma.

**Lemma 8.** *For $\mathbf{x}, \mathbf{z} \in \mathbb{R}^d$ it holds that*

$$\Theta^{(L)}(\mathbf{x}, \mathbf{z}) = r_x r_z \Theta^{(L)}(\hat{\mathbf{x}}, \hat{\mathbf{z}}) = r_x r_z \Theta^{(L)}(\hat{\mathbf{x}}^T \hat{\mathbf{z}}),$$

*where $\Theta^{(L)} = k^{\mathrm{FC}_0(\mathrm{L}+1)}$, as defined in Appendix A.*

*Proof.* We prove this by induction over the recursive definition of $k^{\mathrm{FC}_0(\mathrm{L}+1)} = \Theta^{(L)}(\mathbf{x}, \mathbf{z})$. Let $\mathbf{x}, \mathbf{z} \in \mathbb{R}^d$, then by definition

$$\Theta^{(0)}(\mathbf{x}, \mathbf{z}) = \mathbf{x}^T \mathbf{z} = r_x r_z \Theta^{(0)}(\hat{\mathbf{x}}, \hat{\mathbf{z}}) = r_x r_z \Theta^{(0)}(\hat{\mathbf{x}}^T \hat{\mathbf{z}})$$

and

$$\Sigma^{(0)}(\mathbf{x}, \mathbf{z}) = \mathbf{x}^T \mathbf{z} = r_x r_z \Sigma^{(0)}(\hat{\mathbf{x}}, \hat{\mathbf{z}}) = r_x r_z \Sigma^{(0)}(\hat{\mathbf{x}}^T \mathbf{z})$$

Assuming the induction hypothesis holds for $l$, i.e.,

$$\Theta^{(l)}(\mathbf{x}, \mathbf{z}) = r_x r_z \Theta^{(l)}(\hat{\mathbf{x}}, \hat{\mathbf{z}}) = r_x r_z \Theta^{(l)}(\hat{\mathbf{x}}^T \mathbf{z})$$

and

$$\Sigma^{(l)}(\mathbf{x}, \mathbf{z}) = r_x r_z \Sigma^{(l)}(\hat{\mathbf{x}}, \hat{\mathbf{z}}) = r_x r_z \Sigma^{(l)}(\hat{\mathbf{x}}^T \hat{\mathbf{z}})$$

we prove that those equalities are also true for $l + 1$.

By the definition of $\lambda^{(l)}$ (2) and the induction hypothesis for $\Sigma^{(l)}$ we have that

$$\lambda^{(l)}(\mathbf{x}, \mathbf{z}) = \frac{\Sigma^{(l)}(\mathbf{x}, \mathbf{z})}{\sqrt{\Sigma^{(l)}(\mathbf{x}, \mathbf{x}) \Sigma^{(l)}(\mathbf{z}, \mathbf{z})}} = \frac{\Sigma^{(l)}(\hat{\mathbf{x}}, \hat{\mathbf{z}})}{\sqrt{\Sigma^{(l)}(\hat{\mathbf{x}}_i, \hat{\mathbf{x}}) \Sigma^{(l)}(\hat{\mathbf{z}}, \hat{\mathbf{z}})}} = \lambda^{(l)}(\hat{\mathbf{x}}, \hat{\mathbf{z}}) = \lambda^{(l)}(\hat{\mathbf{x}}^T \hat{\mathbf{z}})$$

Plugging this result in the definitions of $\Sigma$ (3) and $\dot{\Sigma}$ (4), using the induction hypothesis we obtain

$$\begin{aligned}
\Sigma^{(l+1)}(\mathbf{x}, \mathbf{z}) &= r_x r_z \Sigma^{(l+1)}(\hat{\mathbf{x}}, \hat{\mathbf{z}}) = r_x r_z \Sigma^{(l+1)}(\hat{\mathbf{x}}^T \hat{\mathbf{z}}) \\
\dot{\Sigma}^{(l+1)}(\mathbf{x}, \mathbf{z}) &= \dot{\Sigma}^{(l+1)}(\hat{\mathbf{x}}, \hat{\mathbf{z}}) = \dot{\Sigma}^{(l+1)}(\hat{\mathbf{x}}^T \hat{\mathbf{z}})
\end{aligned} \tag{15}$$

Finally, using the recursion formula (1) ($\beta = 0$) and the induction hypothesis for $\Theta^{(l)}$, we obtain

$$\Theta^{(l+1)}(\mathbf{x}, \mathbf{z}) = r_x r_z \Theta^{(l+1)}(\hat{\mathbf{x}}, \hat{\mathbf{z}}) = r_x r_z \Theta^{(l+1)}(\hat{\mathbf{x}}^T \hat{\mathbf{z}})$$

$\square$

A corollary of this Lemma is that $k^{\mathrm{FC}_0(\mathrm{L})}$ is homogeneous of order 1 in $\mathbb{R}^d$, proving the first part of Theorem 4. Also, it is homogeneous of order 0 in $\mathbb{S}^{d-1}$, proving Lemma 7 for $\beta = 0$.

We next turn to proving the second part of Theorem 4, i.e., that $k^{\mathrm{Bias}(\mathrm{L})} = k^{\mathrm{FC}_\beta(\mathrm{L})} - k^{\mathrm{FC}_0(\mathrm{L})}$ is homogeneous of order 0 in $\mathbb{R}^d$. By rewriting the recursive definition of $k^{\mathrm{FC}_\beta(\mathrm{L})}$, shown in Appendix A, we can express $k^{\mathrm{Bias}(\mathrm{L})}$ in the following recursive manner $k^{\mathrm{Bias}(1)} = \beta^2$, and $k^{\mathrm{Bias}(l+1)} = k^{\mathrm{Bias}(l)}\dot{\Sigma} + \beta^2$. Therefore, $k^{\mathrm{Bias}(\mathrm{L})}$ is homogeneous of order zero, since it depends only on $\dot{\Sigma}$, which is by itself homogeneous of order zero (15). This concludes Theorem 4.

Finally, Lemma 7 is proved, since $k^{\mathrm{FC}_\beta(\mathrm{L})} = k^{\mathrm{FC}_0(\mathrm{L})} + k^{\mathrm{Bias}(\mathrm{L})}$, and when restricted to $\mathbb{S}^{d-1}$ both components are homogeneous of order 0.

**Theorem 5.** *Let $p(r)$ be a decaying density on $[0, \infty)$ such that $0 < \int_0^\infty p(r) r^2 dr < \infty$ and $\mathbf{x}, \mathbf{z} \in \mathbb{R}^d$.*

1. *Let $k_0(\mathbf{x}, \mathbf{z})$ be homogeneous of order 1 such that $k_0(\mathbf{x}, \mathbf{z}) = r_x r_z \hat{k}_0(\hat{\mathbf{x}}^T \hat{\mathbf{z}})$. Then its eigenfunctions with respect to $p(r_x)$ are given by $\Psi_{k,j} = a r_x Y_{k,j}(\hat{\mathbf{x}})$, where $Y_{k,j}$ are the spherical harmonics in $\mathbb{S}^{d-1}$ and $a \in \mathbb{R}$.*

2. *Let $\boldsymbol{k}(\mathbf{x}, \mathbf{z}) = \boldsymbol{k}_0(\mathbf{x}, \mathbf{z}) + \boldsymbol{k}_1(\mathbf{x}, \mathbf{z})$ so that $\boldsymbol{k}_0$ as in 1 and $\boldsymbol{k}_1$ is homogeneous of order 0. Then the eigenfunctions of $\boldsymbol{k}$ are of the form $\Psi_{k,j} = (ar_x + b)\, Y_{k,j}\,(\hat{\mathbf{x}})$.*

*Proof.*    1. Since $\hat{\boldsymbol{k}}_0$ is zonal, its Mercer's representation reads

$$\hat{\boldsymbol{k}}_0(\hat{\mathbf{x}}, \hat{\mathbf{z}}) = \sum_{k=0}^{\infty} \lambda_k \sum_{j=1}^{N(d,k)} Y_{k,j}(\hat{\mathbf{x}}) Y_{k,j}(\hat{\mathbf{z}}),$$

where the spherical harmonics $Y_{k,j}$ are the eigenfunctions of $\hat{\boldsymbol{k}}_0$. Consequently, as noted also in [5],

$$\boldsymbol{k}_0(\mathbf{x}, \mathbf{z}) = a^2 \sum_{k=0}^{\infty} \lambda_k \sum_{j=1}^{N(d,k)} r_x Y_{k,j}(\hat{\mathbf{x}}) r_z Y_{k,j}(\hat{\mathbf{z}}).$$

The orthogonality of the eigenfunctions $\Psi_{k,j}(\mathbf{x}) = ar_x Y_{k,j}(\hat{\mathbf{x}})$ is verified as follows. Let $\bar{p}(\mathbf{x})$ denote a probability density on $\mathbb{R}^d$ such that $\bar{p}(\mathbf{x}) = p(r_x)/A(r_x)$, where $A(r_x)$ denotes the surface area of a sphere of radius $r_x$ in $\mathbb{R}^d$. Then,

$$\int_{\mathbb{R}^d} \Psi_{k,j}(\mathbf{x})\Psi_{k',j'}(\mathbf{x})\bar{p}(\mathbf{x})d\mathbf{x} = a^2 \int_0^{\infty} \frac{r_x^{d+1}p(r_x)}{A(r_x)}dr_x \int_{\mathbb{S}^{d-1}} Y_{k,j}(\hat{\mathbf{x}})Y_{k',j'}(\hat{\mathbf{x}})d\hat{\mathbf{x}} = \delta_{k,k'}\delta_{j,j'},$$

where the rightmost equality is due to the orthogonality of the spherical harmonics and by setting

$$a^2 = \left( \int_0^{\infty} \frac{r_x^{d+1}p(r_x)}{A(r_x)}dr_x \right)^{-1}.$$

Clearly this integral is positive, and the conditions of the theorem guarantee that it is finite.

2. By the conditions of the theorem we can write

$$\boldsymbol{k}(\mathbf{x}, \mathbf{z}) = r_x r_z \hat{\boldsymbol{k}}_0(\hat{\mathbf{x}}^T\hat{\mathbf{z}}) + \hat{\boldsymbol{k}}_1(\hat{\mathbf{x}}^T\hat{\mathbf{z}}),$$

where $\hat{\mathbf{x}}, \hat{\mathbf{z}} \in \mathbb{S}^{d-1}$. On the hypersphere the spherical harmonics are the eigenfunctions of $\boldsymbol{k}_0$ and $\boldsymbol{k}_1$. Denote their eigenvalues respectively by $\lambda_k$ and $\mu_k$, so that

$$\int_{\mathbb{S}^{d-1}} \boldsymbol{k}_0(\hat{\mathbf{x}}^T\hat{\mathbf{z}})\bar{Y}_k(\hat{\mathbf{z}})d\hat{\mathbf{z}} = \lambda_k \bar{Y}_k(\hat{\mathbf{x}}) \tag{16}$$

$$\int_{\mathbb{S}^{d-1}} \boldsymbol{k}_1(\hat{\mathbf{x}}^T\hat{\mathbf{z}})\bar{Y}_k(\hat{\mathbf{z}})d\hat{\mathbf{z}} = \mu_k \bar{Y}_k(\hat{\mathbf{x}}), \tag{17}$$

where $\bar{Y}_k(\hat{\mathbf{x}})$ denote the zonal spherical harmonics. We next show that the space spanned by the functions $r_x\bar{Y}_k(\mathbf{x})$ and $\bar{Y}_k(\mathbf{x})$ is fixed under the following integral transform

$$\int_{\mathbb{R}^d} \boldsymbol{k}(\mathbf{x}, \mathbf{z})(\alpha r_z + \beta)\bar{Y}_k(\hat{\mathbf{z}})\bar{p}(\mathbf{z})d\mathbf{z} = (ar_x + b)\bar{Y}_k(\hat{\mathbf{x}}), \tag{18}$$

$\alpha, \beta, a, b \in \mathbb{R}$ are constants. The left hand side can be written as the application of an integral operator $T(\mathbf{x}, \mathbf{z})$ to a function $\Phi_{\alpha,\beta}^k(\mathbf{z}) = (\alpha r_z + \beta)\bar{Y}_k(\hat{\mathbf{z}})$. Expressing this operator application in spherical coordinates yields

$$T(\mathbf{x}, \mathbf{z})\Phi_{\alpha,\beta}^k(\mathbf{z}) = \int_0^{\infty} \frac{p(r_z)r_z^{d-1}}{A(r_z)}dr_z \int_{\hat{\mathbf{z}} \in \mathbb{S}^{d-1}} \left( r_x r_z \boldsymbol{k}_0(\hat{\mathbf{x}}^T\hat{\mathbf{z}}) + \boldsymbol{k}_1(\hat{\mathbf{x}}^T\hat{\mathbf{z}}) \right)(\alpha r_z + \beta)\bar{Y}_k(\hat{\mathbf{z}})d\hat{\mathbf{z}}.$$

We use (16) and (17) to substitute for the inner integral, obtaining

$$T(\mathbf{x}, \mathbf{z})\Phi_{\alpha,\beta}^k(\mathbf{z}) = \int_0^{\infty} \frac{p(r_z)r_z^{d-1}}{A(r_z)}(\lambda_k r_x r_z + \mu_k)(\alpha r_z + \beta)\bar{Y}_k(\hat{\mathbf{x}})dr_z.$$

Together with (18), this can be written as

$$T(\mathbf{x}, \mathbf{z})\Phi_{\alpha,\beta}(\mathbf{z}) = \Phi_{a,b}(\mathbf{x}),$$

where

$$\begin{pmatrix} a \\ b \end{pmatrix} = \begin{pmatrix} \lambda_k & 0 \\ 0 & \mu_k \end{pmatrix} \begin{pmatrix} M_2 & M_1 \\ M_1 & M_0 \end{pmatrix} \begin{pmatrix} \alpha \\ \beta \end{pmatrix}$$

where $M_q = \int_0^\infty \frac{r_z^{q+d-1} p(r_z)}{A(r_z)} dr_z$, $0 \leq q \leq 2$. By the conditions of the theorem these moments are finite. This proves that the space spanned by $\{r_x \bar{Y}(\hat{\mathbf{x}}), \bar{Y}(\hat{\mathbf{x}})\}$ is fixed under $T(\mathbf{x}, \mathbf{z})$, and therefore the eigenfunctions of $\boldsymbol{k}^{\mathrm{FC}_\beta(\mathrm{L})}(\mathbf{x}, \mathbf{z})$ take the form $(\bar{a} r_x + \bar{b}) \bar{Y}(\hat{\mathbf{x}})$ for some constants $\bar{a}, \bar{b}$.

$\square$

The implication of Theorem 5 is that the eigenvectors of $\boldsymbol{k}^{\mathrm{FC}_0(\mathrm{L})}$ are the spherical harmonic functions, scaled by the norm of their arguments. With bias, $\boldsymbol{k}^{\mathrm{FC}_\beta(\mathrm{L})}$ has up to $2N(d, k)$ eigenfunctions for every frequency $k$, of the general form $(a r_x + b) Y_{k,j}(\hat{\mathbf{x}})$ where $a, b$ are constants that differ from one eigenfunction to the next.

## E   Experimental Details

### E.1   The UCI dataSet

In this section, we provide experimental details for the UCI dataset. We use precisely the same pre-processed datasets, and follow the same performance comparison protocol as in [2].

**NTK Specifications**   We reproduced the results of [2] using the publicly available code[1], and followed the same protocol as in [2]. The total number of kernels evaluated in [2] are 15 and the SVM cost value parameter $\mathbf{C}$ is tuned from $10^{-2}$ to $10^4$ by powers of 10. Hence, the total number of hyper-parameter combinations searched using cross-validation is 105 ($15 \times 7$).

**Exponential Kernels Specifications**   For the Laplace and Gaussian kernels, we searched for 10 kernel width values ($1/c$) from $2^{-2} \times \nu$ to $\nu$ in the log space with base 2, where $\nu$ is chosen heuristically as the median of pairwise $l_2$ distances between data points (known as the *median trick* [7]). So, the total number of kernel evaluations is 10. For $\gamma$-exponential, we searched through 5 equally spaced values of $\gamma$ from 0.5 to 2. Since we wanted to keep the number of the kernel evaluations the same as for NTK in [2], we searched through only three kernel bandwidth values ($1/c$) which are 1, $\nu$ and #features (default value in the **sklearn** package[2]). So, the total number of kernel evaluations is 15 ($5 \times 3$).

For a fair comparison with [2], we swept the same range of SVM cost value parameter $\mathbf{C}$ as in [2], i.e., from $10^{-2}$ to $10^4$ by powers of 10. Hence, the total number of hyper-parameter search using cross-validation is 70 ($10 \times 7$) for Laplace and 105 ($15 \times 7$) for $\gamma$-exponential which is the same as for NTK in [2].

### E.2   Large scale datasets

We used the experimental setup mentioned in [14] and the publicly available code[3]. [14] solves kernel ridge regression (KRR [16]) using the FALKON algorithm, which solves the following linear system

$$(K_{nn} + \lambda n I) \, \alpha = \hat{\mathbf{y}},$$

where $K$ is an $n \times n$ kernel matrix defined by $(K)_{ij} = K(x_i, x_j)$, $\hat{\mathbf{y}} = (y_1, \ldots y_n)^T$, and $\lambda$ is the regularization parameter. Refer to [14] for more details.

In Table 1, we provide the hyper parameters chosen with cross validation.

| | MillionSongs [4] | SUSY [13] | HIGGS [13] |
|---|---|---|---|
| H-$\gamma$-exp. | $\gamma = 1.4, \sigma = 5, \lambda = 1e^{-6}$ | $\gamma = 1.8, \sigma = 5, \lambda = 1e^{-7}$ | $\gamma = 1.6, \sigma = 8, \lambda = 1e^{-8}$ |
| H-Laplace | $\sigma = 3, \lambda = 1e^{-6}$ | $\sigma = 4, \lambda = 1e^{-7}$ | $\sigma = 8, \lambda = 1e^{-8}$ |
| NTK | $L = 9, \lambda = 1e^{-9}$ | $L = 3, \lambda = 1e^{-8}$ | $L = 3, \lambda = 1e^{-6}$ |
| H-Gaussian | $\sigma = 8, \lambda = 1e^{-6}$ | $\sigma = 3, \lambda = 1e^{-7}$ | $\sigma = 8, \lambda = 1e^{-8}$ |

Table 1: Hyper-parameters chosen with cross validation for the different kernels.

### E.3   C-Exp: Convolutional Exponential Kernels

Let $\mathbf{x} = (x_1, ..., x_d)^T$ and $\mathbf{z} = (z_1, ..., z_d)^T$ denote two vectorized images. Let $P$ denote a window function (we used $3 \times 3$ windows). Our hierarchical exponential kernels are defined by $\bar{\Theta}(\mathbf{x}, \mathbf{z})$ as follows:

$$
\begin{aligned}
\Theta_{ij}^{[0]}(\mathbf{x}, \mathbf{z}) &= x_i z_j \\
s_{ij}^{[h]}(\mathbf{x}, \mathbf{z}) &= \sum_{m \in P} \Theta^{[h]}(x_{i+m}, z_{j+m}) + \beta^2 \\
\Theta_{ij}^{[h+1]}(\mathbf{x}, \mathbf{z}) &= K(s_{ij}^{[h]}(\mathbf{x}, \mathbf{z}), s_{ii}^{[h]}(\mathbf{x}, \mathbf{x}), s_{jj}^{[h]}(\mathbf{z}, \mathbf{z})) \\
\bar{\Theta}(\mathbf{x}, \mathbf{z}) &= \sum_i \Theta_{ii}^{[L]}(\mathbf{x}, \mathbf{z})
\end{aligned}
$$

where $\beta \geq 0$ denotes the bias and the last step is analogous to a fully connected layer in networks, and we set

$$
K(s_{ij}, s_{ii}, s_{jj}) = \sqrt{s_{ii} s_{jj}}\, \mathbf{k}\left( \frac{s_{ij}}{\sqrt{s_{ii} s_{jj}}} \right)
$$

where $\mathbf{k}$ can be any kernel defined on the sphere. In the experiments we applied this scheme to the three exponential kernels, Laplace, Gaussian and $\gamma$-exponential.

**Technical details**   We used the following four kernels:

**CNTK** [1] $L = 6, \beta = 3$.

**C-Exp Laplace**. $L = 3, \beta = 3, \mathbf{k}(\mathbf{x}^T\mathbf{z}) = a + be^{-c\sqrt{2-2\mathbf{x}^T\mathbf{z}}}$ with $a = -11.491, b = 12.606, c = 0.048$.

**C-Exp $\gamma-$exponential**. $L = 8, \beta = 3, \mathbf{k}(\mathbf{x}^T\mathbf{z}) = a + be^{-c(2-2\mathbf{x}^T\mathbf{z})^{\gamma/2}}$ with $a = -0.276, b = 1.236, c = 0.424, \gamma = 1.888$.

**C-Exp Gaussian**. $L = 12, \beta = 3, \mathbf{k}(\mathbf{x}^T\mathbf{z}) = a + be^{-c(2-2\mathbf{x}^T\mathbf{z})}$ with $a = -0.22, b = 1.166, c = 0.435$.

We set $\beta$ in these experiments with cross validation in $\{1, ..., 10\}$. For each kernel $\mathbf{k}$ above, the parameters $a, b, c$ and $\gamma$ were chosen using non-linear least squares optimization with the objective $\sum_{u \in U}(\mathbf{k}(u) - \mathbf{k}^{\text{FC}_\beta(2)}(u))^2$, where $\mathbf{k}^{\text{FC}_\beta(2)}$ is the NTK for a two-layer network defined in (5) with bias $\beta = 1$, and the set $U$ included (inner products between) pairs of normalized $3 \times 3 \times 3$ patches drawn uniformly from the CIFAR images. The number of layers $L$ is chosen by cross validation.

For the training phase we used 1-hot vectors from which we subtracted 0.1, as in [12]. For the classification phase, as in [10], we normalized the kernel matrices such that all the diagonal elements are ones. To avoid ill conditioned kernel matrices we applied ridge regression with a regularization factor of $\lambda = 5 \cdot 10^{-5}$. Finally, to reduce overall running times, we parallelized the kernel computations on NVIDIA Tesla V100 GPUs.

## Footnotes

[1] `https://github.com/LeoYu/neural-tangent-kernel-UCI`

[2] `https://scikit-learn.org/stable/modules/generated/sklearn.metrics.pairwise.rbf_kernel.html`

[3] `https://github.com/LCSL/FALKON_paper`