[Reviews · NeurIPS 2020]

Review 1

Summary and Contributions: The paper studies properties of neural tangent kernels and relates them to those of the Laplace kernel when restricted to the sphere. In particular, it is shown that the two kernels lead to the same functional space, by having essentially the same eigenvalue decays for the uniform measure on the sphere. The paper provides additional results beyond the sphere, as well as various experiments that highlight the comparable performance of NTK and Laplace kernels, and in many cases improvements in performance by using the gamma-exponential kernel.

Strengths: I find the work quite interesting and significant as it provides a better understanding of the NTK regime for over-parameterized neural networks, and relates the NTK to the more traditional Laplace kernels, which had been empirically observed to behave similarly to deep networks in previous work by Belkin et al. The results on the decay of the Laplace kernel restricted to the sphere are also novel to my knowledge, and of interest by themselves. The experiments are also interesting in that they show the limitations of the NTK compared to usual kernels, in particular that modifications of the standard Laplace kernel tend to outperform the NTK when tuning hyperparameters. The claims are both theoretically and empirically justified, and the paper is clear. I am thus in favor of acceptance.

Weaknesses: I do not see major weaknesses in the paper. Perhaps a minor weakness is that the obtained accuracies on Cifar10 seem quite low, even compared to other works on convolutional kernels. There could also be more discussion on the limitations of the kernel approach/NTK for explaining the success of neural networks, given that the paper shows that the NTK behaves essentially the same as a simple Laplace kernel (this is briefly mentioned in the discussion section, but it seems important to emphasize this from the introduction).

Correctness: yes

Clarity: The paper is well written and pleasant to read.

Relation to Prior Work: yes

Reproducibility: Yes

Additional Feedback: - L118-124: perhaps it should be mentioned here that the work considers the NTK for the ReLU activation? - L201: the meaning of this bound and the following lower bounds could be further clarified. The dependence on the number of examples is also unclear - perhaps standard excess risk bounds for ridge regression would be more appropriate here? - Thm 5: Can something also be said about how the eigenvalues are affected compared to the spherical case? **** update after rebuttal **** Thanks to the authors for their response. One minor detail to add is that it would be useful to include a few more experimental details in the main paper, as opposed to in the appendix (e.g. hyperparameter selection in the convolutional case).


Review 2

Summary and Contributions: The paper theoretically proves that NTK corresponding to a two-layer FC neural network with bias has the same RKHS of Laplace Kernel for normalized data on hyperspheres. The paper also proves the RKHS of Laplace Kernel is a subset of RKHS of NTK corresponding to multi-layer FC NN, and conjecture these two are also the same by experimental evidence. The experiments show that the Laplace kernel performs similarly to NTK in three settings.

Strengths: - The theory of the paper is sound to me. The result is novel and interesting. - The experiments are mostly complete and follow standard cross-validation.

Weaknesses: - The theoretical contribution of this paper is questionable. The paper mentioned similar results in the paper but doesn't give a comparison of the proof techniques, especially for Therorem 1. - For the experiment on convolutional kernels, the author doesn't mention why they need additional parameters a, b, c. It makes me feel like the authors are just try to manually approximate NTK better by adding these parameters. The authors also didn't mention how they choose the hyperparameter beta.

Correctness: I didn't find flaws in the proof, and the experiments are mostly sound.

Clarity: The paper is mostly well-written. Minor comments: - line 108, should use \langle and \rangle for inner product. - A better definition of RKHS is required. - line 141, probably need to define or explain spherical harmonics.

Relation to Prior Work: The paper have listed previous works. I suggest a more detailed comparison to those works.

Reproducibility: Yes

Additional Feedback: Can you clarify why you need a, b, c in the experiment on convolutional kernels? And how is beta chosen? ----- post author response ---- The authors addressed most of my concerns, but I believe choosing a, b, c to explicitly fit NTK is not a good approach for C-exp. You should modify the way you choose a, b, c and re-run the experiments in Sec 4.3.


Review 3

Summary and Contributions: This paper derives new results for the eigenvalues/eigenfunctions of the Neural Tangent Kernel for fully-connected networks (deeper networks, with bias), as well as the Laplace kernel, and demonstrates the similarity between the two kernels both theoretically and empirically. More specifically: --Theoretical proof that NTK eigenvalues, when data lies uniformly on the hypersphere in d-dimensions, decay like O(k^-d) for networks with one-hidden layer and no faster than O(k^-d) for deeper networks [Theorem 1]. Laplace kernels in the same setting decay also as O(k^-d) [Theorem 2]. Hence there is an equivalence between the RKHS of the two kernels, which is also larger than that of the Gaussian kernel [Theorem 3]. --Results for the eigenfunctions of the fully-connected NTK in R^d. --Empirical comparison of the performance of NTK (fully-connected network), Laplace, Gaussian, and \gamma-exponential kernels (as well as homogeneous versions where applicable) on the UCI dataset as well as larger datasets. Indeed, the NTK kernels perform similarly to Laplace. --Empirical comparison, on CIFAR-10, between convolutional NTK (CNTK) and "convolutional" versions of the {Laplace, Gaussian, \gamma-exponential} kernels obtained by recursive application of these kernels to image patches. Interestingly, these latter kernels can perform similarly to CNTK.

Strengths: Soundness of the claims: The results are clearly stated, seem to be theoretically well-supported, and are also confirmed empirically (numerical comparison of the spectra, angular dependence in Fig. 1, as well as indirectly via the performance comparisons in Section 4). Significance, relevance: I think the results in the paper will be of interest to the community studying overparameterized networks as well those working on kernel methods. Naively, one might think that the new class of Neural Tangent Kernels, by virtue of being connected to the compositional/recursive nature of fully-connected networks, would be a richer kernel class than previously known. This paper refutes this for the particular class of NTKs connected to fully-connected architectures. It also performs an early investigation touching on how to construct a powerful class of convolutional-like kernels from existing primitives (\gamma-exponential kernels) by working with image patches. I think the paper also does a nice job balancing theory and empirical work, and I appreciate that the authors did performance comparisons in Section 4 for the variety of kernels they discuss.

Weaknesses: Some treatment (which could be purely empirical) of non-uniform data on the sphere or in R^d would enhance the paper a bit, as it is closer to a realistic setting. (As far as I could gather, the distributions considered are all uniform.) For instance, how do the NTK and Laplace kernel eigenvalues/eigenfunctions compare empirically for the CIFAR-10 dataset? (The performance comparisons are suggestive but only indirect.) Additionally, the main text seems to not discuss which activation functions are being treated, and the supplementary seems to treat the case of Relu. If that is the case, this should be mentioned early on in the main text and abstract; also, some discussion (empirical or theoretical) of any changes in the main conclusions for alternative common activations would be beneficial. Additionally, the discussion of related works could be improved (see "Relation to prior work").

Correctness: The claims and methods appear to be correct, though I have not checked in detail the theory proofs in the supplementary.

Clarity: The paper is clearly written.

Relation to Prior Work: Ref. 4 and 5 in the bibliography are repeated. Some of the related works are not accurately cited and should be corrected. Ref. 41, 35 treat Bayesian inference and do not discuss last layer training under gradient descent. The correspondence with GPs in Ref. 35 was done for single hidden layer fully-connected networks only. The papers https://arxiv.org/abs/1711.00165 and https://arxiv.org/abs/1804.11271 (both in ICLR 2018) derived the GP correspondence for deep fully-connected networks. Ref. 17 derived the arccosine kernel but did not discuss GP correspondences. I'm also not certain that Ref. 1 in its published form (note the arxiv analog spans several different versions) discussed the NTK.

Reproducibility: Yes

Additional Feedback:


Review 4

Summary and Contributions: new connections shown between neural tangent kernel and Laplace kernel and improved results with gamma exponential kernel

Strengths: - new connections shown at the theoretical level and through empirical evaluation - excellent relevance to NeurIPS

Weaknesses: - see below for suggestions for improvement

Correctness: - see below - kernel regression (2) does not correspond with the empirical section

Clarity: overall well written

Relation to Prior Work: can be improved

Reproducibility: Yes

Additional Feedback: Studying the connection between neural networks and kernel methods is important and interesting in general. The authors make a nice contribution in this direction. - in the abstract it is stated that the same eigenfunctions and eigenvalues lead to the same function classes. Later in the paper this is somewhat explained, but it would be good to give additional references on this from learning theory. It is only based on [26] which is more related to approximation theory. - section 2: other early work on the connection between SVM and neural networks is: Cortes C,, Vapnik V., Support vector networks, Machine Learning 1995 (with the use of tanh kernel) Suykens J., Vandewalle J., Training multilayer perceptron classifiers based on a modified support vector method'', IEEE Transactions on Neural Networks, 1999 (defining the hidden layer as feature map in the primal) - eq (2) is rather an interpolation problem. This is not the standard kernel regression approach which allows errors with use of a loss function. It is also not corresponding to the kernel ridge regression used in the experiments. - Lemma 1: please explain what a zonal kernel is and give reference(s). - l.188: convexity versus non-convexity is overlooked here. Please comment on this. - broader impact: this remains to be completed. Possibly the authors may comment here on interpretability of models - A l.17 why is the notation \dot{Sigma} used? Is the dot related to a derivative? - l.49 slightly superior -> slightly better Thanks to authors for the replies, I have taken it into consideration.

[Author Response · NeurIPS 2020]

We thank the reviewers for their encouraging reviews and detailed comments, and are pleased that they all recommend
the paper be accepted.

**R2** *Theoretical contribution and proof technique.* We would like to emphasize our contributions: (1) Deriving the
eigenvalue decay rate of NTK with one hidden layer and bias and a lower bound for deeper networks for data distributed
uniformly on the hypersphere (Thm. 1). (2) Deriving eigenvalue decay under the same conditions for the Laplace kernel
(Thm. 2). (3) Theoretical comparison of the class of functions (RKHS) that correspond to NTK (deep and shallow),
Laplace and Gaussian kernels (Thm. 3). (4) Results on the eigenfunctions of NTK outside the sphere (Thm. 5). (5)
Empirical comparison of NTK to exponential kernels on various standard datasets. (6) Empirical comparison of CNTK
and convolutional versions of exponential kernels.

Regarding our proof techniques, the proof in Thm. 1 for NTK with two layers and bias borrows techniques from [6].
Our proof technique for deep networks uses the algebra of RKHSs and is therefore novel in this context. Our proof of
Thm. 2 derives bounds that result from the relation between the Fourier expansion of the Laplace kernel in $\mathbb{R}^d$ to its
harmonic expansion in $\mathbb{S}^{d-1}$. Finally, Thm. 5 derives the eigenfunctions of NTK in $\mathbb{R}^d$ by using invariant properties of
NTK (established in Thm. 4) and identifying the spaces fixed under the appropriate integral transform.

**R2** *"why they need additional parameters a, b, c."* The hyperparameter $c$ is used commonly to determine the sharpness
of the Laplace kernel. We note that analogously NTK becomes sharper for deeper networks. Controlling $c$ is therefore
analogous to depth selection. The remaining parameters apply an affine modulation to the Laplace kernel. In regression
scaling a kernel has no effect, while an affine shift only affects the DC component. Therefore $a$ and $b$ have only
little effect on the results of regression. They are however important in repeated application of the kernel, such as in
the convolutional C-Exp algorithm. The bias term $\beta$ in the C-Exp experiments is chosen through cross validation in
$\{1, ..., 10\}$.

**R3** *"Some treatment of non-uniform data."* Analyzing the similarity between the kernels under nonuniform distributions
is a worthwhile future objective. Figure 1 below indicates that both the eigenvalues and eigenfunctions of NTK and the
Laplace kernel closely match also for nonuniform distributions. This is shown for a piecewise uniform density with
three bins of ratios 5:12:1 in $\mathbb{S}^1$ (left and middle panels). The right panel shows the eigenvalues obtained for the UCI
Abalone dataset.

Figure 1: An overlay of the eigenvalues (left, log-log plot) and example eigenfunctions (middle) of NTK and the Laplace kernels
for data in $\mathbb{S}^1$. Right: eigenvalues on the UCI Abalone dataset.

**R1** and **R3** *"Which activation functions are being treated."* Our paper indeed focuses on the ReLU function only – we
apologize for the unintended omission. With different activation functions the RKHS of NTK would differ, depending
on the degree of smoothness of the function. For example, the eigenvalues for NTK with tanh activation appears to
decay exponentially fast.

**R1** *"the obtained accuracies on Cifar10 seem quite low."* Our experiments are run under the same conditions as in [2].
Incorporating an average pooling layer is likely to lead to better accuracies, as is demonstrated for CNTK in [28].

**R1** and **R4** *"The dependence on the number of examples is also unclear."* This generalization result depends on the
mesh norm, which captures the minimal distance between training points. Alternative bounds, such as in [10] page 9,
provide such dependence.

**R1**: *Eigenvalues in $\mathbb{R}^d$.* Compared to $\mathbb{S}^{d-1}$, the eigenvalues are scaled uniformly, depending on the radial distribution.

**R4**: *"Convexity versus non-convexity is overlooked here."* Solving kernel ridge regression using the Representer theorem
involves least squares minimization, which is convex. Since the kernel matrix is positive definite we can use a faster,
primal-dual formulation [16], where both are convex.

We will modify the paper to improve the discussion of previous work, change Eq. (2) to include ridge regression, and
address the rest of the reviewers' comments.

[Meta-Review · NeurIPS 2020]

Four knowledgeable referees rate this article 7,6,7,6. The article shows theoretically (and experimentally) that the NTK closely resembles the Laplace kernel. The reviewers agree that the results should be of interest to the community, but also have made various suggestions on how the article could be improved. R2 agrees with R1 that the theoretical results are novel and interesting. However, that the experiments are missing details, and hence the moderate enthusiasm in the rating, but would be more positive if the authors would re-run convolutional kernel experiments. R3 agrees with R2 that important aspects of the experiments should not be buried in the appendix. R4 considers that the paper contributes valuable novel insights but considers that the technical quality could be higher. I am recommending accept, but I would like to encourage the authors to address the recommendations and suggestions from the referees in the preparation of the final manuscript.